# A Permian fish reveals widespread distribution of neopterygian-like jaw suspension

**Thodoris Argyriou[1,2,3,4]\*[†‡], Sam Giles[5,6], Matt Friedman[6,7]**

[1]Paleontological Institute and Museum, University of Zurich, Zurich, Switzerland; [2]CR2P, MNHN-CNRS-Sorbonne Universitée, Muséum National d'Histoire Naturelle, Paris, France; [3]Department of Earth and Environmental Sciences, Paleontology & Geobiology, Ludwig-Maximilians-Universität München, München, Germany; [4]GeoBio-Center, Ludwig-Maximilians-Universität München, München, Germany; [5]School of Geography, Earth and Environmental Sciences, University of Birmingham, Birmingham, United Kingdom; [6]Scientific Associate, The Natural History Museum, London, United Kingdom; [7]Museum of Paleontology and Department of Earth and Environmental Sciences, University of Michigan, Ann Arbor, United States

**\*For correspondence:**
t.argyriou@lrz.uni-muenchen.de

**Present address:** [†]Department of Earth and Environmental Sciences, Paleontology & Geobiology, Ludwig-Maximilians-Universität München, München, Germany; [‡]GeoBio-Center, Ludwig-Maximilians-Universität München, München, Germany

**Competing interest:** The authors declare that no competing interests exist.

**Abstract** The actinopterygian crown group (comprising all living ray-finned fishes) originated by the end of the Carboniferous. However, most late Paleozoic taxa are stem actinopterygians, and broadly resemble stratigraphically older taxa. The early Permian †*Brachydegma caelatum* is notable for its three-dimensional preservation and past phylogenetic interpretations as a nested member of the neopterygian crown. Here, we use computed microtomography to redescribe †*Brachydegma*, uncovering an unanticipated combination of primitive (e.g., aortic canal; immobile maxilla) and derived (e.g., differentiated occipital ossifications; posterior stem of parasphenoid; two accessory hyoidean ossifications; double jaw joint) dermal and endoskeletal traits relative to most other Paleozoic actinopterygians. Some of these features were previously thought to be restricted to the neopterygian crown. The precise phylogenetic position of †*Brachydegma* is unclear, with placements either on the polypterid stem or as an early-diverging stem neopterygian. However, our analyses decisively reject previous placements of †*Brachydegma* in the neopterygian crown. Critically, we demonstrate that key endoskeletal components of the hyoid portion of the suspensorium of crown neopterygians appeared deeper in the tree than previously thought.

## Editor's evaluation

This work is a valuable description and analysis of a fossil taxon with importance for understanding the complexities of actinopterygian evolution and is relevant to biologists and paleontologists interested in actinopterygians. Although in many ways *Brachydegma* raises more questions than it answers, the manuscript represents a substantive contribution to the subject – especially as *Brachydegma* has been thought by some researchers to occupy a quite different phylogenetic position and thus have a different significance to the overall story.

## Introduction

Living ray-finned fishes (Actinopterygii) include three lineages: the early diverging Cladistia (bichirs and reedfish, 14 spp.), the Chondrostei (sturgeons and paddlefishes, 27 spp.), and the markedly speciose Neopterygii (Holostei [gars and bowfin], 8 spp. + Teleostei, ~32,000 spp.) (*Nelson et al., 2016*).

Molecular and fossil evidence suggests that these lineages diverged in the Devonian–Carboniferous interval, with an early Carboniferous divergence age estimate being more likely (*Broughton et al., 2013*; *Near et al., 2013*; *Giles et al., 2017*). A number of key characters supporting relationships among major living actinopterygian groups relate to internal parts of the skeleton, and ambiguities in the relationships of some extinct lineages to these extant radiations might reflect limited information on endoskeletal traits in fossils. The majority of phylogenetic analyses incorporating extant and Palaeozoic-Mesozoic ray-fins have traditionally recovered cladistians, the extant sister lineage to all other living actinopterygians, as branching deep within a Devonian radiation (*Patterson, 1982*; *Gardiner, 1984*; *Long, 1988*; *Gardiner and Schaeffer, 1989*; *Gardiner et al., 1996*; *Coates, 1999*; *Gardiner et al., 2005*; *Xu et al., 2014*, but see *Cloutier and Arratia, 2004*; *Mickle et al., 2009*). More recently, the hypothesis of †scanilepiforms as stem-cladistians (*Giles et al., 2017*) has led to a major revision of early ray-fin relationships, with the notable result that almost all Devonian-Permian and many Triassic taxa fall on the actinopterygian stem. Notwithstanding analyses that recover cladistians as a deep Devonian radiation, surprisingly few Paleozoic taxa (e.g., †*Platysomus* [*Moy-Thomas and Dyne, 1938*], †eurynotiforms [*Sallan and Coates, 2013*; *Friedman et al., 2018*], †*Discoserra* [*Lund, 2000*; *Hurley et al., 2007*], †*Ebenaqua* [*Campbell and Phuoc, 1983*], †*Acentrophorus* [*Gill, 1923*; *Gardiner, 1960*; *Patterson, 1973*]) have been resolved or verbally placed within the actinopterygian crown (*Cloutier and Arratia, 2004*; *Hurley et al., 2007*; *Xu et al., 2014*; *Giles et al., 2017*; *Argyriou et al., 2018*; *Latimer and Giles, 2018*), and knowledge of the endoskeleton of these taxa is rudimentary at best.

Features of the hyoid arch bear on the relationships of living actinopterygian lineages (*Patterson, 1973*; *Patterson, 1982*; *Véran, 1988*; *Gardiner et al., 1996*). Each extant lineage has a distinctive geometry and arrangement of the hyoid skeleton, with major differences relating to the size and number of elements between the dorsal and ventral components of the arch. The presence of a single element linking the dorsal (hyomandibula) and ventral (ceratohyal) components characterizes cladistians (*Allis, 1922*; *Jollie, 1984*; *Claeson et al., 2007*). Two intermediate elements are present in chondrosteans (*Grande and Bemis, 1991*; *Hilton et al., 2011*) and neopterygians (*Patterson, 1973*; *Grande and Bemis, 1998*; *Grande, 2010*; *Arratia, 2013*). In halecomorphs, these ossifications are arranged in a sub-parallel manner, with one of these articulating with the lower jaw and forming the so-called double jaw joint (*Patterson, 1973*; *Patterson, 1982*; *Grande and Bemis, 1998*). A single accessory element has been described for the majority of Devonian-Triassic non-neopterygian actinopterygians (e.g., †*Mimipiscis Gardiner, 1984*; †*Coccocephalichthys Poplin and Véran, 1996*; †*Pteronisculus Nielsen, 1942*; †*Australosomus Nielsen, 1949*; †*Gogosardina Choo et al., 2009*). However, *Véran, 1988*, indicated the presence of two elements in some Triassic-Jurassic non-neopterygian taxa, such as †*Boreosomus reuterskioldi* and †*Ptycholepis bollensis*. Two intermediary elements, geometrically arranged in a manner similar to that of halecomorphs, are unambiguously present in the Early Triassic †parasemionotids (*Patterson, 1973*; *Patterson, 1982*; *Olsen, 1984*; *Gardiner et al., 1996*; but see *Arratia, 2013*, for a possible similar geometry in early teleosts), as well as younger extinct groups, like †pycnodonts (*Nursall and Maisey, 1987*; *Gardiner et al., 1996*; *Kriwet, 2005*). These patterns have inspired two interpretive models for the evolution and homology of hyoid arch elements. *Patterson, 1982*, proposed that a single intermediate element represented the primitive actinopterygian condition, but this was subsequently disputed by *Véran, 1988*, who countered that two elements are plesiomorphic for the group. There are a number of challenges to distinguishing between these hypotheses, not least the difficulty in interpreting incomplete and poorly preserved fossils, a lack of detailed descriptions for articulated and in situ fossil hyoid arches, and also the varying degrees of mineralization of these elements in vivo. Accessory hyoid elements represent a key source of anatomical support for actinopterygian relationships, but there is a profound lack of information for these features in all but a handful of Paleozoic and early Mesozoic taxa (*Patterson, 1973*; *Patterson, 1982*; *Olsen, 1984*; *Véran, 1988*; *Gardiner et al., 1996*; *Gardiner et al., 2005*).

Previous research on the endoskeletal anatomy of fossil actinopterygians has mostly focused on generalized Devonian-Carboniferous forms (*Poplin, 1974*; *Gardiner, 1984*; *Poplin and Véran, 1996*; *Coates, 1999*; *Hamel and Poplin, 2008*; *Giles et al., 2015a*; *Giles et al., 2015b*; *Pradel et al., 2016*), or both generalized and anatomically specialized Triassic taxa (*Nielsen, 1942*; *Nielsen, 1949*; *Patterson, 1975*; *Olsen, 1984*; *Giles et al., 2017*; *Argyriou et al., 2018*; *Latimer and Giles, 2018*). The Permian is an important link between the stem actinopterygian dominated Devonian-Carboniferous

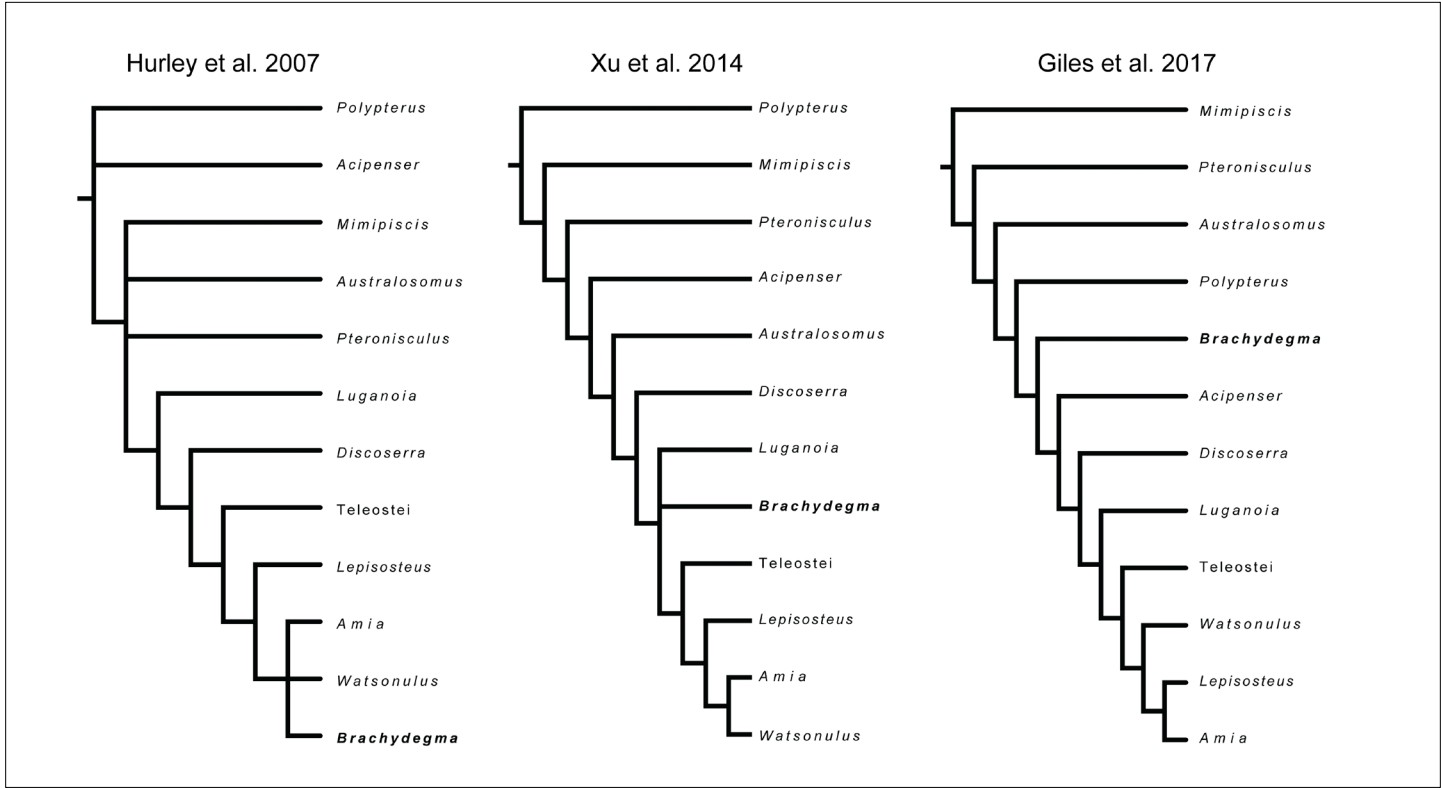

**Figure 1.** Previously hypothesized phylogenetic placements of †*Brachydegma caelatum*. Simplified trees given contain an indicative subset of taxa common to all published phylogenetic hypotheses.

and the neopterygian-rich faunas of Triassic and younger strata. However, the Permian also represents a major knowledge gap for all aspects of actinopterygian biology (*Hurley et al., 2007*; *Friedman and Sallan, 2012*; *Sallan, 2014*), including endoskeletal structure (for partially preserved examples, see *Aldinger, 1937*; *Dunkle, 1946*; *Jessen, 1972*; *Figueroa et al., 2019*).

†*Brachydegma caelatum* is one of the few Permian ray-fins represented by three-dimensional cranial material, and constitutes a key taxon in debates on patterns and timing of major divergences within actinopterygian phylogeny (*Hurley et al., 2007*; *Near et al., 2012*; *Broughton et al., 2013*; *Xu et al., 2014*; *Giles et al., 2017*; summarized in *Figure 1*). Known only from two specimens (*Figures 2–4*; *Appendix 1—figure 1*) from the Cisuralian (early Permian) Red Beds of Texas, TX, USA (*Dunkle, 1939*), †*Brachydegma* has been previously interpreted exclusively through external examination of the type specimen, leading to radically divergent phylogenetic interpretations (*Figure 1*). †*Brachydegma* was initially aligned with anatomically generalized groups of uncertain monophyly that likely represent stem actinopterygians (†elonichthyids *Dunkle, 1939*, or †acrolepidids *Schaeffer, 1973*), but a later reappraisal identified it as a halecomorph, predating previous fossil-based minima for the age of the neopterygian crown and the split between holostean and teleostean lineages by roughly 30 Ma (*Hurley et al., 2007*). Subsequent assessments (*Figure 1*) challenged the halecomorph (*Near et al., 2012*; *Xu et al., 2014*) or even total-group neopterygian (*Broughton et al., 2013*; *Giles et al., 2017*) affinities of †*Brachydegma*. None, however, provided new anatomical data for the specimens.

Here, we use computed microtomography (μCT) to reveal, for the first time, the character-rich anatomy of the braincase, mandibular and hyoid arches, branchial skeleton, pectoral girdle, and the anterior portion of the axial skeleton of †*Brachydegma* (*Appendix 1—figure 2*), with the goal of informing the phylogenetic position of this enigmatic taxon. We find that, unlike other known Palaeozoic ray-fins, the internal anatomy of †*Brachydegma* bears a number of unexpected specializations and character combinations. Critically, the hyoid arch anatomy of †*Brachydegma* indicates a more complicated evolution of accessory hyoid elements and their involvement in jaw suspension than is currently appreciated. Moreover, †*Brachydegma* presents the first – almost complete and largely

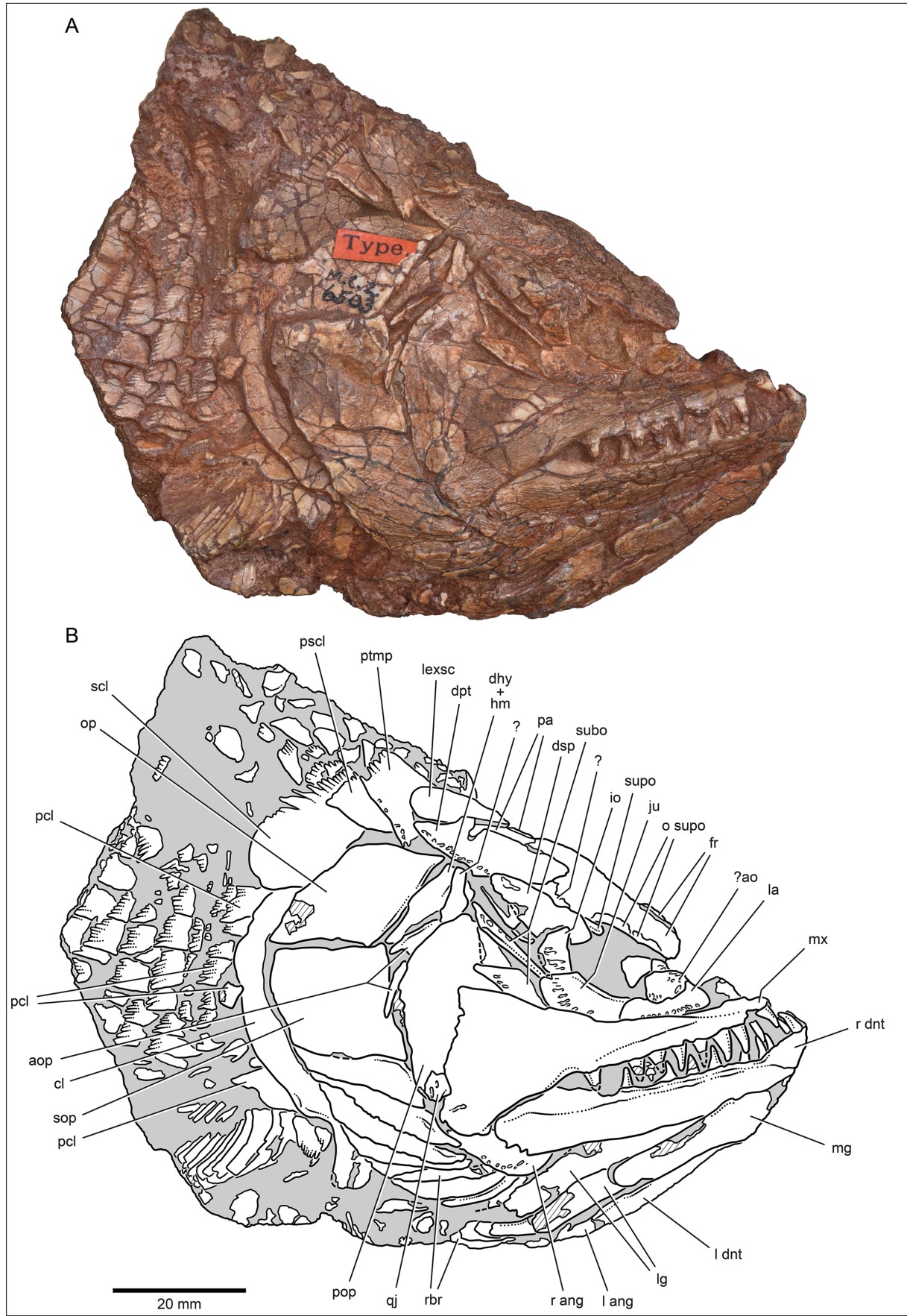

articulated – model of branchial anatomy in a Permian actinopterygian, as well as a rare example of pectoral and axial endoskeletal structure from the latter part of the Paleozoic.

# Results

## Systematic paleontology

> Actinopterygii sensu Goodrich 1930
> †*Brachydegma caelatum* Dunkle 1939

## Material

Museum of Comparative Zoology (Harvard University) MCZ VPF-6503, †*Brachydegma caelatum*, holotype, slightly compressed laterally, preserving cranial and anterior postcranial skeleton; MCZ VPF-6504, †*Brachydegma caelatum*, paratype, dorsoventrally compressed, with dermal skeleton eroded away, but preserving internal cranial and aspects of anterior postcranial skeleton.

## Locality and geological background

Both specimens of †*Brachydegma* come from the stratigraphically oldest deposits of the northern part of the Clear Fork Formation (formerly recognized as the Arroyo Formation; see *Nelson et al., 2013*), Indian Creek, Baylor County, TX (*Dunkle, 1939*). The Clear Fork formation has been biostratigraphically dated to the North American 'Leonardian' Stage, which largely overlaps with the late Kungurian (stage range: 283.5–273.01 Ma)–early Roadian (stage range: 273.01–266.9 Ma) interval of the late Cisuralian (*Nelson et al., 2013*). The Clear Fork Formation is characterized by ferruginous, calcitic-sandy, terrigenous facies (*Dunkle, 1939*; *Olson, 1989*; *Nelson et al., 2013*), broadly assigned to coastal floodplain environments (*Nelson et al., 2013*). The accompanying vertebrate fauna includes †xenacanths, lungfishes, and tetrapods, including the †pelycosaur †*Dimetrodon* (*Olson, 1989*).

## Revised diagnosis

Actinopterygian characterized by the following unique combination of characters: occiput comprising three separate ossifications; absence of a dermal basipterygoid process; parasphenoid reaching posterior to the ventral otic fissure; lateral dorsal aortae extending along the ventral surface of parasphenoid; immobile maxilla in broad connection with the palate; coronoid process absent or greatly reduced; at least three suborbitals; at least two 'accessory opercles' below dermohyal; independently ossified symplectic and interhyal with sub-parallel arrangement.

## Exoskeletal cranial anatomy

We provide a redescription of †*Brachydegma caelatum*, mostly based on the better-preserved type specimen MCZ VPF-6503. We present detailed photographs and illustrations (*Figures 2–4*). For previous interpretations, see *Dunkle, 1939*, and *Hurley et al., 2007*. We only refer to the paratype MCZ VPF-6504 when it shows features absent from the type specimen. Our anatomical interpretations are in broad agreement with the original description (*Dunkle, 1939*), and are largely based on direct observations of the fossils, since superficial ossifications were not possible to reconstruct from the µCT data (*Appendix 1—figure 2*). Many parts of the external skeleton are badly fractured or preserved. This is particularly the case for the rostral area of the holotype, where individual bones have been

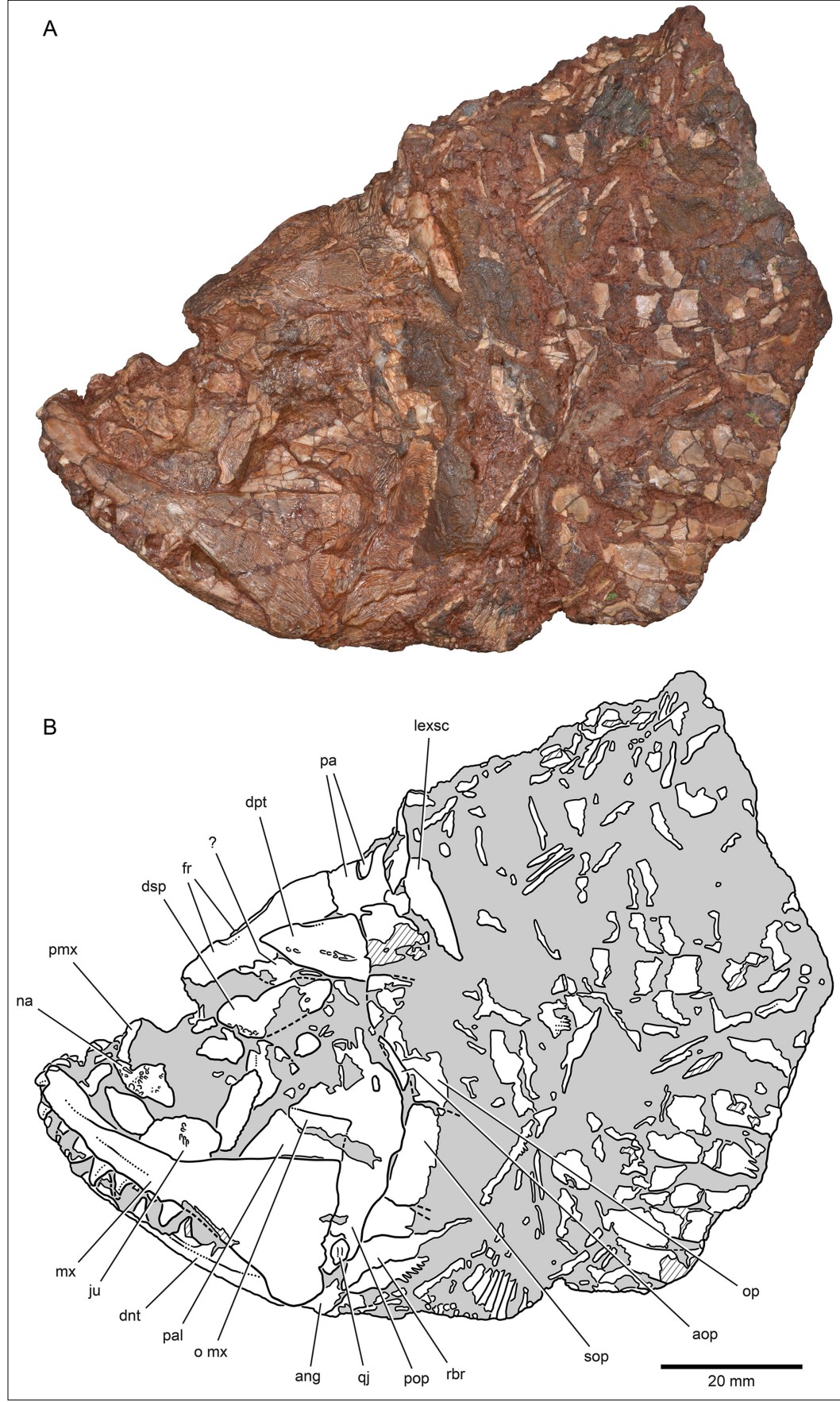

Figure 3 continued on next page

*Figure 3 continued*

**Figure 3.** External anatomy of †*Brachydegma caelatum* holotype (MCZ VPF-6503), left lateral view. Specimen photograph (**A**) and weighted-line drawing (**B**). Abbreviations: **?**, uncertain; **ang**, angular; **aop**, accessory opercle; **dnt**, dentary; **dpt**, dermopterotic; **dsp**, dermosphenotic; **fr**, frontal; **ju**, jugal; **lexsc**, lateral extrascapular; **mx**, maxilla; **na**, nasal; **o mx**, overlap area on the preopercle for the maxilla; **op**, opercle; **pa**, parietal; **pal**, palate; **pmx**, premaxilla; **pop**, preopercle; **qj**, quadratojugal; **rbr**, branchiostegal ray; **sop**, subopercle. Dashed lines indicate a missing margin; hatching indicates a broken surface; gray tone indicates matrix. Scale bar equals 20 mm.

subject to varying interpretations by past authors. The dermal ossifications of the skull are externally ornamented with densely packed and often anastomosing vermiform ridges.

*Skull roof*: The rostrum and the anterior part of the skull roof are incompletely preserved. A fragmentary bone bearing a conical tooth represents a fragment of the likely paired premaxilla. The frontals are longer than wide, with their posteroventral margin bearing an indentation for the insertion of the dermopterotics. The parietals are quadrangular, and the midline suture between the bilateral counterparts of the frontals and parietals anastomoses. Each parietal bears a lateral extension that inserts into the body of the adjacent dermopterotic. There is no independent accessory parietal as suggested by *Dunkle, 1939*. The dermopterotics are longer than wide. The left dermopterotic appears divided into two parts (dpt, *Figures 2–4*), but this reflects a combination of breakage and the lateral process of the associated parietal. On the right side of the skull, the anterior and posterior portions of the dermopterotic are clearly connected by a ventral bridge of bone and unambiguously constitute a single ossification, in contrast to the separate intertemporal and supratemporal of many Devonian and some Carboniferous actinopterygians (*Gardiner, 1984*; *Gardiner and Schaeffer, 1989*). A pair of lateromedially elongate lateral extrascapulars lie posterior to the parietals.

*Circumorbital, cheek, and operculogular ossifications*: The canal-bearing dermosphenotic (postorbital in *Dunkle, 1939*, and *Hurley et al., 2007*) is sub-rectangular, with a posterior ramus. Its anterior end is fragmented, but it likely did not reach far anteriorly above the orbit. We recognize a triangular area of bone on both sides of the skull, framed by the dermopterotic posteriorly, the frontal dorsomedially, and the dermosphenotic ventrolaterally. This region was interpreted as a dermosphenotic by *Dunkle, 1939*, and *Hurley et al., 2007*, but it shows no obvious signs of pores for a sensory canal. Three infraorbitals (named here as infraorbital, jugal, lachrymal) surround the posterior and ventral margins of the orbit, with the lachrymal bearing a possible anterior thickening. The dislocated canal-bearing element situated immediately dorsal to the lachrymal on the right side of the type specimen is a putative antorbital. Another, more elongate canal-bearing bone is present on the left side of the specimen in association with the tooth-bearing fragment of the premaxilla. Previously identified as an antorbital (*Hurley et al., 2007*), it is best identified as a nasal (na, *Figures 3–4*) based on the presence of an ascending arm and a possible narial notch. Another possible nasal, or alternatively a fragment of the rostral shield, is present in the paratype, but in a very poorly preserved state (*Appendix 1—figure 1*). At least two supraorbitals must have been present, as evidenced by corresponding sockets on the right frontal. At least three anamestic suborbitals arranged in a dorsoventral series separate the infraorbitals from the preopercle. A supramaxilla is absent. The preopercle is taller than wide, and sits almost upright in the cheek. It bears an overlap area for the maxilla, shown clearly on the right side of the skull where the two bones have pulled away from each other. The pronounced ventral limb of the preopercle connects to a small quadratojugal. Three additional dorsoventrally arranged anamestic bones separate the preopercle from the opercular series, on the right side of the type specimen (*Figure 2*). The dorsal-most constitutes the unfused dermohyal, and the ventral two represent accessory opercles. The latter are broadly comparable to those of the †acrolepidids sensu lato (*Aldinger, 1937*; *Nielsen, 1942*). The opercle is rhomboidal and is of almost equal size to the more quadrate subopercle. At least eight branchiostegal rays are present. The two lateral gulars are rostrocaudally elongate and underlie the posterior half of the lower jaw. The median gular is longer than wide, and of subequal length to the lateral gulars.

*Shoulder girdle*: The dermal skeleton of the pectoral girdle is largely preserved on the right side of the type specimen. The posttemporal is subquadrate and seems to form an anterolateral ramus, likely excluding the extrascapular from the lateral margin of the skull roof. An additional dermal ossification

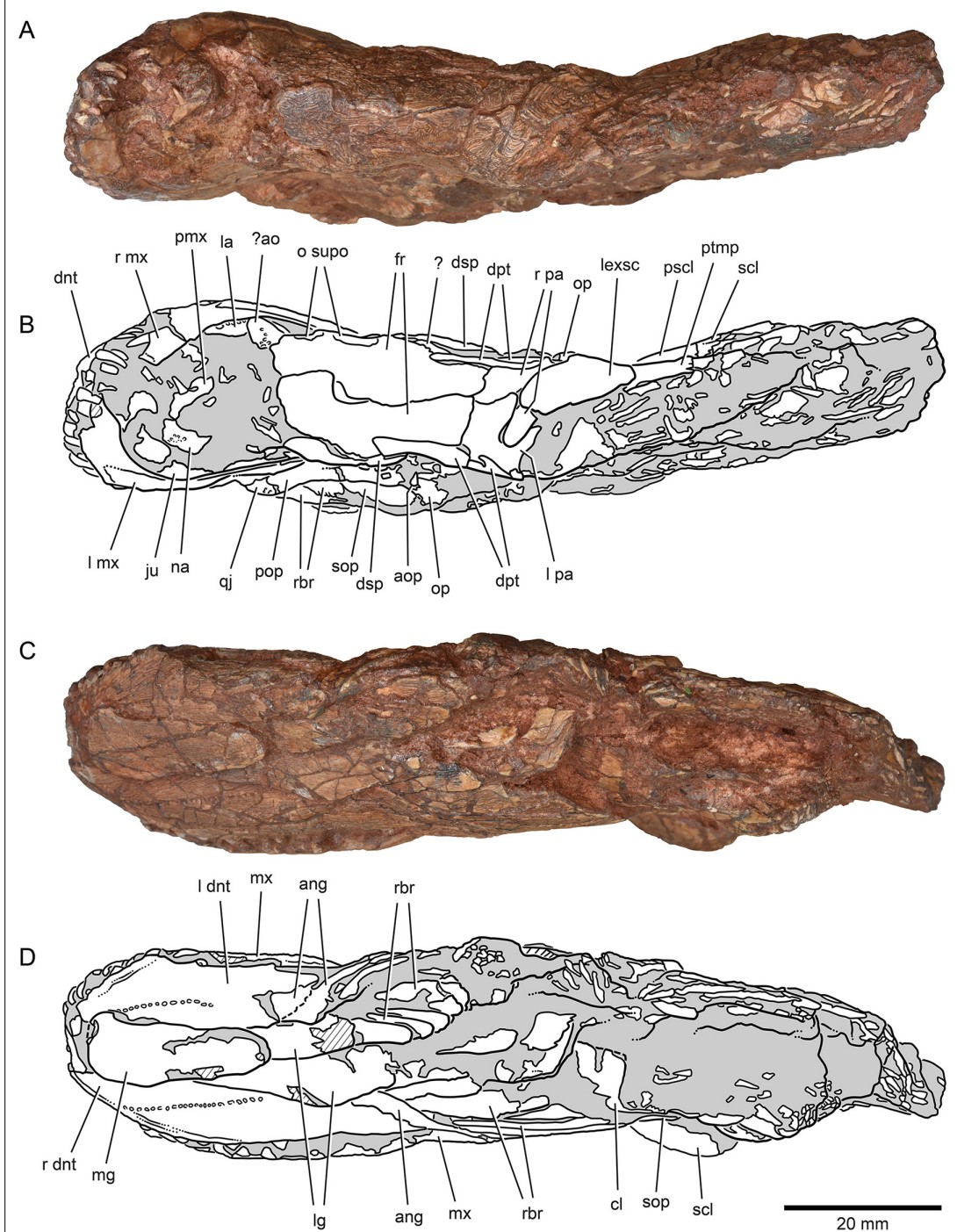

**Figure 4.** External anatomy of †*Brachydegma caelatum* holotype (MCZ VPF-6503), dorsal and ventral views. Specimen photograph (**A**) and weighted-line drawing (**B**) in dorsal view. Specimen photograph (**C**) and weighted-line drawing (**D**) in ventral view. Abbreviations: **?**, uncertain; **?ao**, antorbital; **ang**, angular; **aop**, accessory opercle; **cl**, cleithrum; **dnt**, dentary; **dpt**, dermopterotic; **fr**, frontal; **ju**, jugal; **la**, lachrymal; **lexsc**, lateral extrascapular; **lg**, lateral gular; **mg**, median gular; **mx**, maxilla; **na**, nasal; **o supo**, overlap areas on frontal for supraorbitals; **op**, opercle; **pa**, parietal; **pmx**, premaxilla; **pop**, preopercle; **pscl**, presupracleithrum; **ptmp**, posttemporal; **qj**, quadratojugal; **rbr**, branchiostegal rays; **scl**, supracleithrum; **sop**, subopercle. Abbreviations preceded by l or r indicate left or right structure, respectively. Dashed lines indicate a missing margin; hatching indicates a broken surface; gray tone indicates matrix. Scale bar equals 20 mm.

lies ventral to the posttemporal, but is partially obscured by it. We identify this triangular ossification as a presupracleithrum. The supracleithrum is ovoidal and larger than the posttemporal, reaches further posteriorly than the cleithrum, and forms a strongly convex and serrated posterior margin. The postcleithra are poorly preserved, but appear to have been three to four in number. This series includes the 'accessory' postcleithrum of *Hurley et al., 2007*. Fringing fulcra line the anterior margin of the pectoral fin.

## Endoskeletal anatomy

*Braincase and parasphenoid*: The braincase comprises several ossifications, with large gaps presumably filled by cartilage in life. Three distinct ossifications contribute to the occipital region (boc, exo, *Figure 5*): a basioccipital and a pair of exoccipitals, comparable to neopterygians (*Patterson, 1975*; *Grande and Bemis, 1998*; *Grande, 2010*). The anterior margin of the occiput is well demarcated, likely indicating an unmineralized oticooccipital fissure. The basioccipital projects posteriorly and encloses a short endoskeletal aortic canal. The exoccipitals form the dorsal margin of the notochordal opening and surround the foramen magnum. Lateral to the foramen magnum, the exoccipitals expand posteriorly, and anteriorly flare laterally, forming possible craniospinal processes. Unlike in many crown neopterygians (*Patterson, 1975*; *Grande and Bemis, 1998*; *Grande, 2010*), the exoccipitals do not enclose the vagus nerve. The otic capsules are poorly preserved and lie within an area of low contrast in both specimens, rendering them impossible to interpret from the scans. The sphenoid ossification is partially preserved in the paratype, and exhibits a deep, paired posterior myodome (osph, pmy, *Figure 5A, F and I*). The interorbital septum forms as a thick median pillar. There is no median optic foramen, with the optic nerves entering each orbit separately. Identification of accessory nerve and venous foramina is difficult in both specimens. It is thus not possible to confidently determine whether features such as spiracular canals or endoskeletal basipterygoid processes were present.

The parasphenoid (psp, *Figure 5*) underlies most of the braincase, extending far posterior to the ventral fissure but terminating before reaching the back of the braincase. A similar condition is present in all crown actinopterygians (*Allis, 1922*; *Patterson, 1975*; *Olsen, 1984*; *Hilton et al., 2011*; *Giles et al., 2017*), †saurichthyids (*Argyriou et al., 2018*), and several Paleozoic forms of uncertain affinity (e.g., †platysomids [*Moy-Thomas and Dyne, 1938*], †eurynotiforms [(*Friedman et al., 2018*], and †*Sphaerolepis* [*Stamberg, 1991*]). The lateral dorsal aortae of †*Brachydegma* exit the basicranium and extend along the ventral surface of the parasphenoid, divided by a median keel. The posterodorsally directed ascending processes are well developed and bear a spiracular groove. A dermal basipterygoid process is absent. The parasphenoid appears to be edentulous, but this may be an artifact of preservation or lack of contrast. The presence of a buccohypophyseal canal could not be ascertained. The anterior process of the parasphenoid, anterior to the orbitosphenoid region of the braincase, exhibits a posterodorsally-anteroventrally directed groove on each side (apal, *Figure 5A, C, F, G and H*). These grooves either transmitted the parabasal canals or were employed in the articulation of the palate, as in, for example, polypterids and sarcopterygians (*Lemberg et al., 2021*).

*Palate and associated ossifications*: The dermal and endoskeletal palate of †*Brachydegma* (*Figure 6A–E, H and J*) is deep along most of its length, with a convex, imperforate dorsal margin. Processes for articulation with the braincase and parasphenoid are not apparent. Ventrally, the palate forms a broad flange that abuts the prominent medial shelf of the maxilla (mxhl, vpl, *Figure 6C–E and K*). This indicates a strong connection between the two, rendering maxillary kinesis impossible. A reinforced lateral palatal (ectopterygoid) process forms the anterior border of the adductor notch of the palate, and abuts the maxilla. Dermal palatal bones are difficult to separate in tomograms, but appear to comprise multiple ossifications. A long and broad accessory vomer lies along the medial surface of the anterior half of the palate. The quadrate is located posteroventrally and bears two small articular condyles and a gentle posterior groove. A dorsolateral flange on the palate, slightly anterior to the level of the jaw articulation, is the only trace of the metapterygoid, and seems to resemble the metapterygoid process of neopterygians (*Olsen, 1984*; *Arratia and Schultze, 1991*). An autopalatine has not been located in either specimen and was probably not mineralized in life. An ossified labial element is present near the ventral opening of the mandibular adductor chamber, in both sides of the type specimen (calg, lbe, *Figure 6A, F and G*). The one of the right side is more completely preserved, and is approximately tear-shaped and perforated, resembling the ones found in, for example, †*Boreosomus reuterskioldi* and †*Ptycholepis bollensis* (*Véran, 1996*).

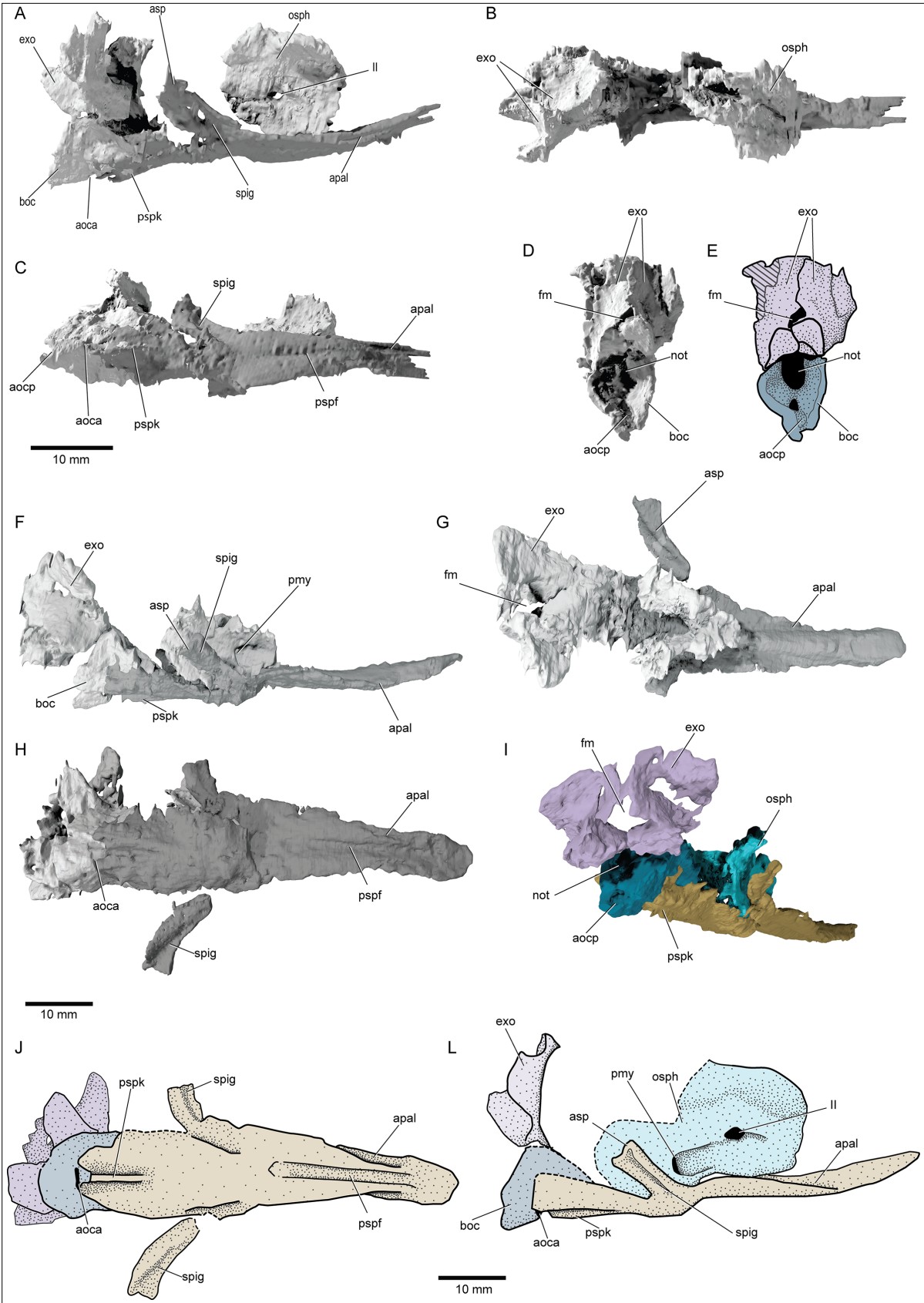

**Figure 5.** Braincase and parasphenoid of †*Brachydegma caelatum*. Type specimen (MCZ VPF-6503) in (**A**) right lateral, (**B**) dorsal, (**C**) ventral, (**D**) posterior views; (**E**) line drawing of D with separate ossifications color coded. Paratype (MCZ VPF-3504) in (**F**) right lateral, (**G**) dorsal, (**H**) ventral,
*Figure 5 continued on next page*

*Figure 5 continued*

(**I**) posterolateral view with separate ossifications color coded; (**J**) line drawing of H with separate ossifications color coded, (**L**) composite reconstruction of preserved aspects of the braincase in lateral view with separate ossifications color coded. Abbreviations: **II**, orbital opening; **aoca**, anterior opening or aortic canal; **aocp**, posterior opening of aortic canal; **apal**, furrows for suspension of palate or parabasal canals; **asp**, ascending process of parasphenoid; **boc**, basioccipital; **exo**, exoccipitals; **fm**, foramen magnum; **not**, notochordal opening; **osph**, orbitosphenoid; **pmy**, posterior myodome; **pspf**, median furrow of parasphenoid; **pspk**, ventral keel of parasphenoid; **spig**, spiracular groove. Dashed lines indicate a missing margin. Scale bars equal 10 mm.

*Hyoid arch*: The hyomandibula (*Figure 7A–D*) forms rather broad and distinct dorsal and ventral limbs, is perforate, and appears to bear a short and broad opercular process. An unfused dermohyal sits on the lateral surface of the dorsal limb of the hyomandibula. On the left side of the holotype, two accessory hyoid elements are preserved, in close association with the ventral tip of the hyomandibula (ih, sy, *Figure 8A–E*), and are aligned sub-parallel to one another. We identify the anterior ossification as a symplectic and the posterior ossification as an interhyal based on their position, morphology, and relationship with other ossifications (see Discussion). The symplectic is subquadrangular, and articulates with the anteroventral tip of the hyomandibula (syf, *Figure 8B*). Its anterior surface forms a keel, which likely fit in a groove on the posterior surface of the quadrate (qdgr, *Figure 8B*). The anteroventral tip of the symplectic forms a thickening, or condyle, which inserts in a facet on the posterior surface of the articular; its posteroventral tip is produced as a thin, ventrally directed process, which contacts the posterior surface of the articular (cnd, vpsy?, *Figure 8C–E*). A faint groove on the anterodorsal face is likely for the passage of the afferent mandibular artery (afmd, *Figure 8C–E*). A similar groove is present in some †parasemionotids (NHMD 74424A; *Figure 8F–I*), and the putative symplectic of †*Pteronisculus* (*Figure 8J–M*). The presence of this feature in the latter is congruent with previous observations made by Véran on the putative symplectics of stem actinopterygians (*Véran, 1988*). The more posterior of the two elements present in †*Brachydegma*, identified as an interhyal, is rod-shaped, and articulates with the posteroventral tip of the hyomandibula. Only the more robust, anterior ossification (symplectic) is apparent on the left side of the type specimen (*Figure 7A, B*). The region between the hyomandibula and ceratohyal is poorly preserved in the paratype and thus the presence of accessory elements cannot be determined (*Appendix 1—figure 2F*). The contact between the symplectic and articular – a double jaw joint – as well as the sub-parallel arrangement of the symplectic and the interhyal resemble that of crown neopterygians, such as †parasemionotids (*Stensiö, 1932*; *Patterson, 1973*; *Olsen, 1984*; *Figure 8F–I*), †*Parapholidophorus* (*Arratia, 2013*), as well as †pycnodonts (*Nursall and Maisey, 1987*; *Gardiner et al., 1996*; *Kriwet, 2005*). A single, laterally grooved, plate-like ceratohyal is ossified on each side of the hyoid arch of †*Brachydegma* (*Figures 7B, E and 9A, B*).

*Branchial skeleton*: The branchial skeleton of the type specimen is nearly complete (*Figure 9*), although the pharyngobranchials are somewhat disarticulated and shifted from their life positions. We have attempted to reconstruct the branchial series, but the identification and positioning of the suprapharyngobranchials remains somewhat speculative. The branchial skeleton of †*Brachydegma* exhibits the common motif of five ossified branchial arches. One or two basibranchials are preserved (bb/bh?, bb, *Figure 9C, E*). The anteriormost and smallest of the two exhibits a subtriangular cross section; it could alternatively constitute a dislocated basihyal, similar to that of, for example, †*Pteronisculus* (*Nielsen, 1942*). The second (or only) basibranchial is subtriangular in cross section, and exhibits a flat dorsal surface. This element is constricted at mid-length. Five ceratobranchials are present (cb1–5, *Figure 9*). The first four ceratobranchials are curved. Their posteroventral surface is grooved, whereas anterodorsally they accommodate a series of small, multicuspid rakers. The fifth ceratobranchial is reduced to a tiny rod-like structure. The dorsal bones of the gill arches are partially disarticulated. The first three epibranchials (ep1–3, *Figure 9*) bear uncinate processes, with the first two being particularly well developed, like in, for example, †*Australosomus* (*Nielsen, 1949*). The uncinate processes of the second and third epibranchials are oriented medially. The fourth epibranchial is short and wide and forms a long and thin anterior process, and a laterally expanded plate supporting the passage of the efferent branchial artery. No expanded toothplates are associated with the epibranchials. The first infrapharyngobranchial is rod-shaped and edentulous. The element tentatively identified as the first suprapharyngobranchial is hooked, possibly engulfing its corresponding

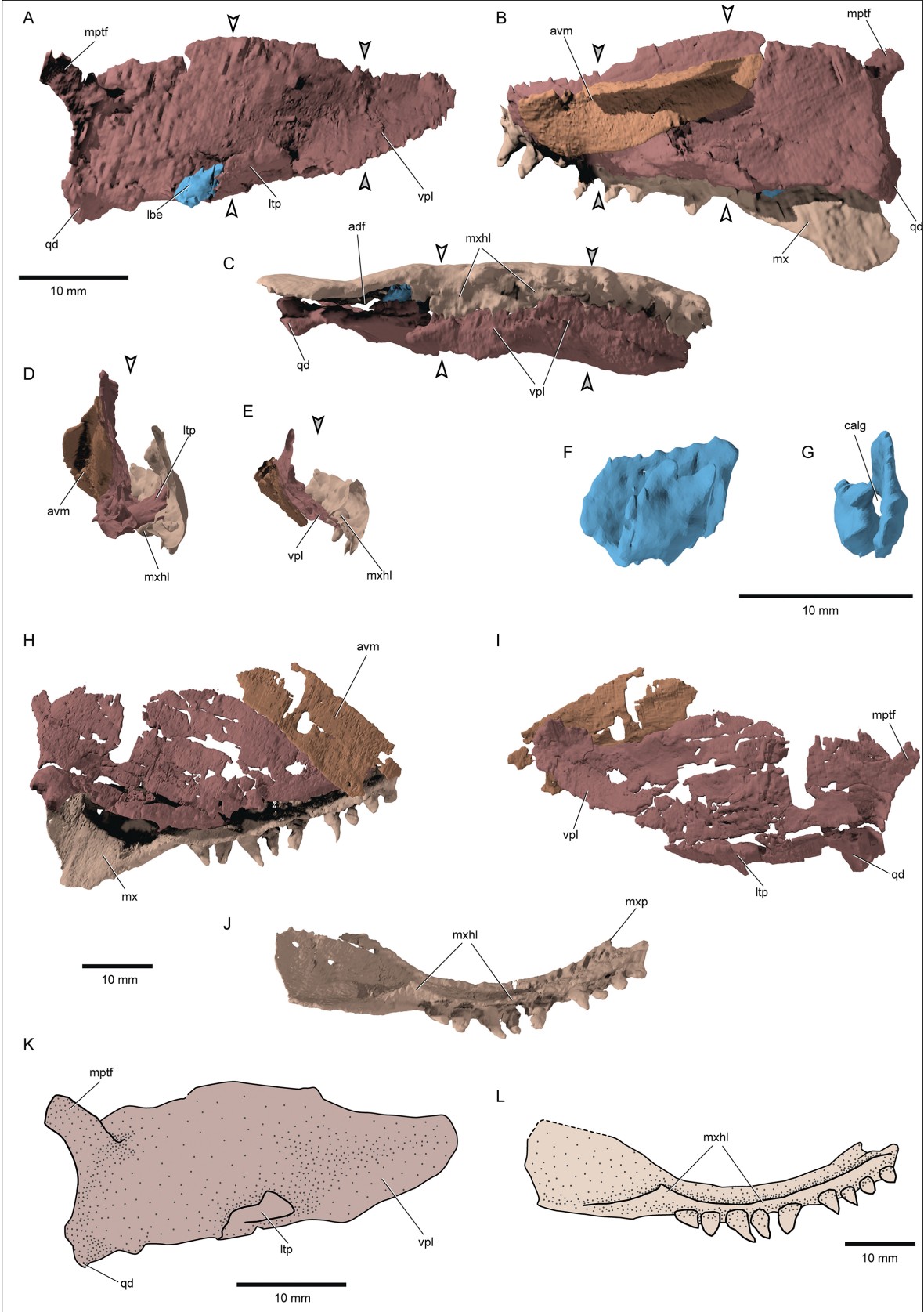

**Figure 6.** Palatal complex and maxilla of †*Brachydegma caelatum.* Type specimen (MCZ VPF-6503): (**A**) Palatal complex and 'labial element' in lateral view; palatal complex and maxilla in medial (**B**) and (**C**) ventral views; cross sections demarcated by arrows in A, B, C showing the relationship of palatal

*Figure 6 continued on next page*

*Figure 6 continued*

complex and maxilla at (**D**) the level of lateral process and (**E**) the level of the orbital notch. (**F**) Lateral view of the ossified labial element; (**G**) anterior view of the same element. Paratype (MCZ VPF-6504): (**H**) palatal complex and maxilla in medial view; (**I**) palatal complex in lateral view; (**J**) maxilla in ventromedial view. (**L**) Schematic reconstruction of palatal complex in lateral view. (**M**) Schematic reconstruction of maxilla in ventromedial view. Abbreviations: **avm**, accessory vomer; **calg**, canal for the passage of ligaments; **lbe**, ossified 'labial element'; **ltp**, lateral process of the ectopterygoid; **mptf**, metapterygoideal flange; **mx**, maxilla; **mxhl**, horizontal lamina of maxilla; **mxp**, ethmoid articulation of maxilla; **qd**, quadrate; **vpl**, ventrolateral palatal lamina. Scale bars equal 10 mm.

efferent arterial vessel. The second infrapharyngobranchial is wider and plate-like, and bears putative teeth on its ventral surface. The second suprapharyngobranchial is laterally compressed and forms a weakly forked distal proximal margin, which likely aided in its suspension from the occipital region. An additional rod-like element associated with the third branchial arch is tentatively identified as an ossified third infrapharyngobranchial.

*Jaws*: The maxilla exhibits a robust horizontal process (*Figure 6J*), which supports a single series of large, pointed teeth. The posterior plate of the maxilla is well developed, flat, and tall, with its posterodorsal margin fitting in a notch on the preopercle. On the left side of the specimen (*Figure 2C*), the maxilla is slightly disarticulated from the preopercle, which likely gave the impression of maxillary kinesis (hereby deemed absent) to previous authors (*Hurley et al., 2007*). Medially, the maxilla forms a well-developed horizontal lamina for attachment to the dermal palate (mxhl, *Figure 6C–E and J*), similar to that of stem actinopterygians like †*Mimipiscis*, †*Pteronisculus*, or †*Australosomus* (*Nielsen, 1942*; *Nielsen, 1949*; *Gardiner, 1984*). The maxilla of MCZ VPF-6504 does not possess a posterior notch, contrasting a previous reconstruction of this feature (*Hurley et al., 2007*). We do not find evidence for a rod-like articular process with the ethmoid region of the skull like that of neopterygians (see, for example, *Amia* in Grande and Bemis, 1998). Instead, a short and thin, plate-like anterior process for firm articulation with the ethmoid and dermal snout ossifications of the skull is present (mxp, *Figure 6J*). The dentary (dnt, *Figures 2–4* and *Figure 7A, F–H*) is the principal bone of the lateral surface of the jaw, and bears a single row of teeth. The prearticular occupies most of the mesial surface of the lower jaw, but no teeth are apparent in the tomograms. Coronoids cannot be distinguished from the prearticular. Both the external and the μCT-aided examinations of the holotype suggest the presence of a surangular (sang, *Figure 7F*) on the posterodorsal corner of the jaw and an angular on the posteroventral, but sutures between these bones could not be reliably determined. There is no well-developed coronoid process, but a faint one at best formed by the surangular alone. This process – if accepted as such – clearly differs from that of deeply diverging crown neopterygians, which receives contributions by the dentary and the prearticular (*Olsen, 1984*; *Grande and Bemis, 1998*; *Figure 8F*). What was previously reconstructed as a pronounced coronoid process on the type specimen (*Hurley et al., 2007*) corresponds to a smooth, gentle shelf of the dentary for overlap by the maxilla (mxs, *Figure 7F*). The deep adductor fossa is surrounded by the articular posteriorly, the prearticular medially, and the surangular and dentary laterally. In addition to the two depressions for the quadrate, the articular bears a posterior flat facet for the insertion of the condyle of the symplectic (syf, *Figure 7G, H*).

*Postcranial skeleton*: The pectoral girdle (*Figure 10A–G*) is largely preserved. The clavicles are broad triangular plates covering the anterior process of the massive cleithra, resembling the primitive (*Nielsen, 1942*; *Gardiner, 1984*) condition associated with non-neopterygians (*Jollie, 1984*; *Hilton et al., 2011*) but retained in †parasemionotids (*Olsen, 1984*) and some stem teleosts (*Arratia, 2013*). The cleithra lack the well-developed postbranchial lamina of many deeply diverging actinopterygians like †*Trawdenia* (*Coates and Tietjen, 2018*), but also extant chondrosteans (*Hilton et al., 2011*) and the parasemionotid †*Watsonulus* (*Olsen, 1984*). The cleithrum is rather tall and forms an acute dorsal tip. The posterior notch of the cleithrum faces posteroventrally. The scapulocoracoids of the type specimen could be largely reconstructed (scc, *Figure 10C–G*), revealing a peculiar set of characters. The dorsal (scapular) portion of the scapulocoracoid is well developed on both sides, whereas the ventral (coracoid) portion is not preserved and was conceivably cartilaginous. This is supported by the presence of a subhorizontal facet (fcpl, *Figure 10D*) on the mesial surface of the middle region of the scapulocoracoid. The scapulocoracoid is attached to the cleithrum by means of a broad dorsal plate,

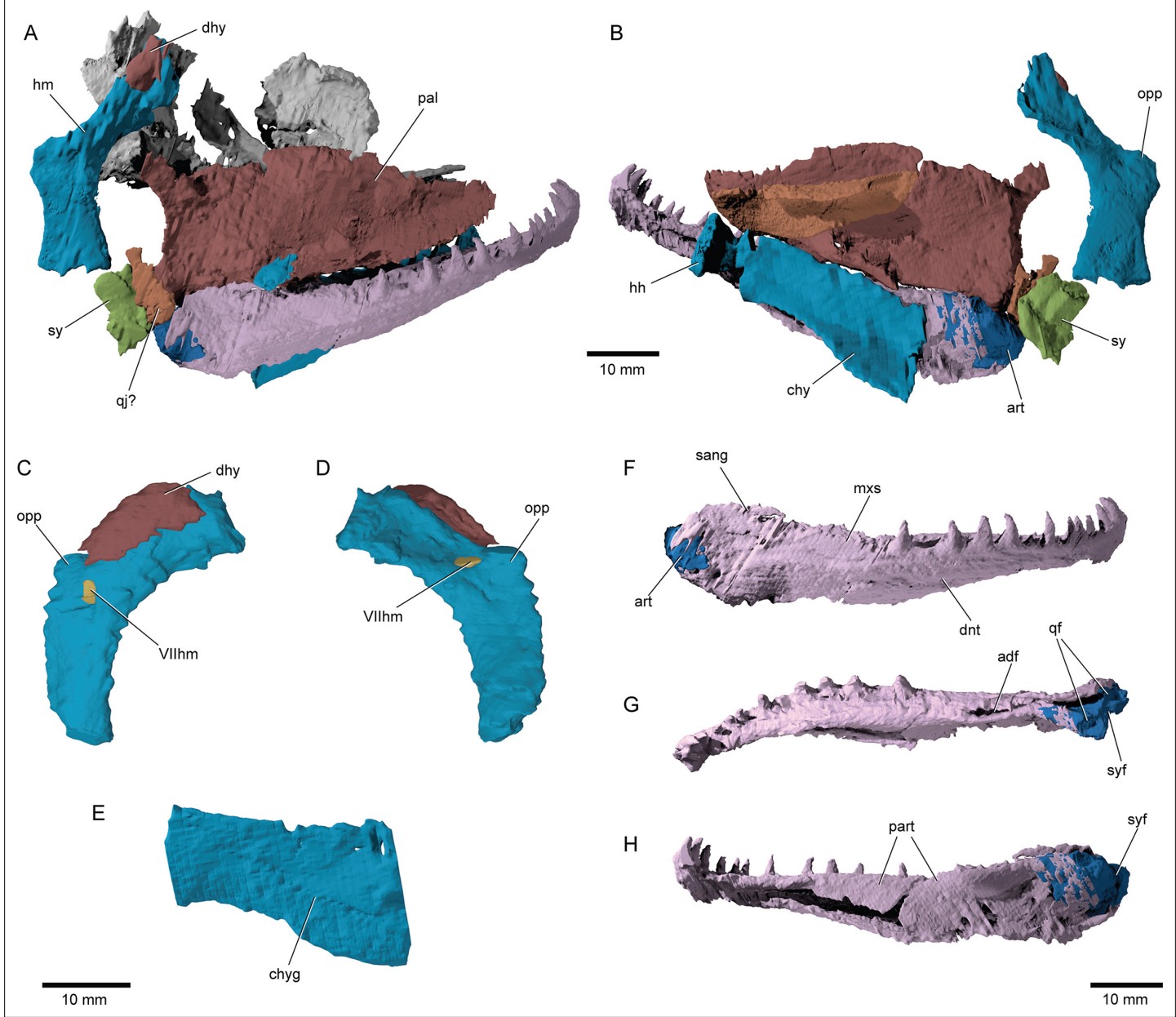

**Figure 7.** Suspensorium and lower jaw of †*Brachydegma caelatum*. Right palatal complex and suspensorium of type specimen (MCZ VPF-6503) shown as preserved in (**A**) lateral and (**B**) medial views. Paratype (MCZ VPF-6504): right hyomandibula in (**C**) lateral and (**D**) medial views; (**E**) left ceratohyal in lateral view. Right lower jaw of type specimen in (**F**) lateral, (**G**) dorsal, and (**H**) medial views. Abbreviations: **VIIhm**, hyomandibular trunk of facial nerve; **adf**, adductor fossa; **art**, articular; **chy**, ceratohyal; **chyg**, arterial groove; **dhy**, dermohyal; **dnt**, dentary; **hh**, hypohyal; **hm**, hyomandibula; **mxs**, maxillary shelf on dentary; **opp**, opercular process; **pal**, palatal complex; **qf**, quadrate facets; **qj?**, putative quadratojugal; **sang**, surangular; **sy**, symplectic; **syf**, symplectic fossa. Scale bars equal 10 mm.

and forms a dorsal, medially hooked, protrusion, which corresponds to the dorsal end of the meso-coracoid arch (dmca, *Figure 10D*). The remainder of the mesocoracoid arch is not present and may have been cartilaginous. Sockets on the ventromedial side of the 'scapular' portion of the bone likely received either the ventral tip of a cartilaginous mesocoracoid arch and/or the putatively cartilaginous coracoid portion of the bone (vmca, *Figure 10D*). A dorsal scapular process is not preserved. The supracoracoid foramen is oval and directed laterally-lateroventrally. It is preceded by a smaller round foramen. The middle posterior region for the articulation of the radials is medioventrally directed,

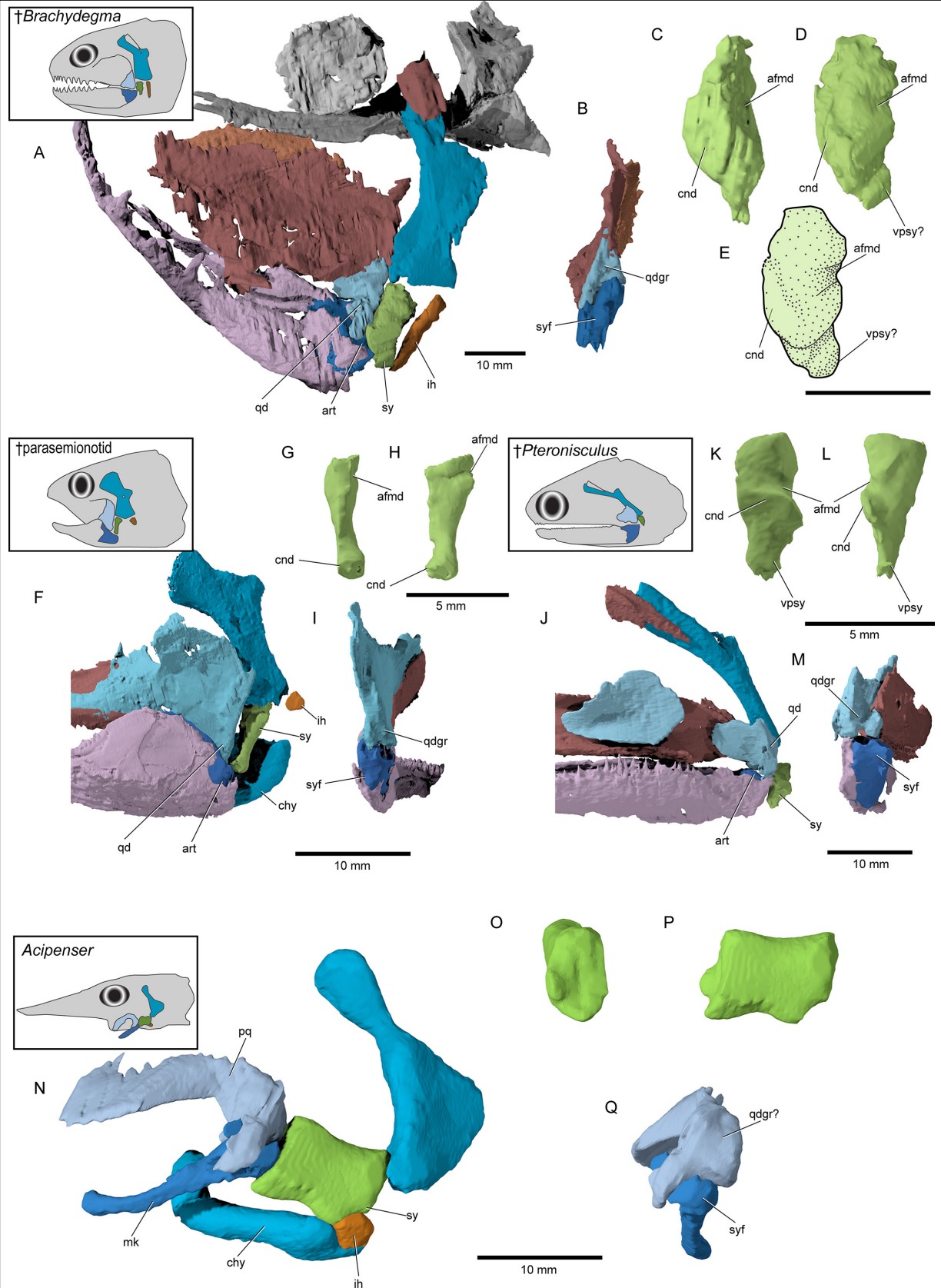

**Figure 8.** Accessory hyoidean ossifications in †*Brachydegma caelatum* and other actinopterygians. †*Brachydegma caelatum* type specimen (MCZ VPF-6503): (**A**) left suspensorium shown as preserved in lateral view; (**B**) posterior view of quadrate and articular; (**C**) anterior and (**D**) anterolateral detail of *Figure 8 continued on next page*

*Figure 8 continued*

symplectic; (**E**) line drawing of D. †Parasemionotidae indet. (NHMD 74424A): (**F**) left suspensorium shown as preserved in lateral view; in (**G**) anterior and (**H**) lateral detail of symplectic; (**I**) posterior view of quadrate and articular; †*Pteronisculus gunnari* (NHMD 73588A): (**J**) right suspensorium in mirrored lateral view; (**K**) anterior and (**L**) lateral detail of symplectic; (**M**) posterior view of quadrate and articular; *Acipenser brevirostrum* (UMMZ 64250): (**N**) left suspensorium in lateral view; (**O**) anterior and (**P**) lateral detail of 'symplectic'; (**Q**) posterior view of palatoquadrate and mackel's cartilage. Abbreviations: **afmd**, groove for afferent mandibular artery; **art**, articular; **chy**, ceratohyal; **cnd**, anterior condyle of symplectic; **ih**, interhyal; **mk**, Meckel's cartilage; **pq**, palatoquadrate; **qd**, quadrate; **qdgr**, posterior groove on quadrate; **sy**, symplectic; **syf**, symplectic fossa; **vpsy**, ventral process of symplectic. Scale bars for A, B, F, I, J, M–Q equal 10 mm; scale bars for C–E, G, H, K, L equal 5 mm.

as in most fossil and recent actinopterygians (*Jessen, 1972*). A single series of rod-like radials lie sub-parallel to each other. A short, stocky element is tentatively identified as a propterygium (ppt?, *Figure 10F*), but we cannot confirm whether it is perforate or not.

The notochord is unconstricted, and only arcual elements are apparent (arc, bd, bv, *Figure 10H–J*). Supraneurals are not observed in the anterior vertebral segments. Dorsally, the first abdominal vertebral segment comprises a stout, paired basidorsal bearing a short neural spine and a prezygapophysis. More posterior basidorsals exhibit thinner but longer neural spines and thinner prezygapophyses. A faint transverse canal extends along the medial surface of each neural hemispine. As in most crown actinopterygians, excluding teleosts (*Arratia, 2013*), epineural processes are not developed. A median hemi-cylindrical element in one of the anteriormost vertebral segments lacks parapophyses and might have resulted from the fusion of two basiventrals. However, all remaining basiventrals are paired and bear short, ventrolaterally expanding parapophyses, though no ossified ribs are present. The rhomboid scales (*Figure 10K*) of †*Brachydegma* exhibit a dorsal articular peg and a small anterodorsal process. Their posterior scale margin forms acute serrations.

## Phylogenetic results

*Equally weighted parsimony*: Our equally weighted parsimony analysis recovered 1412 most parsimonious trees of 1652 steps (*Figure 11*). The crown neopterygian clade receives low Bremer support in our analysis (Bremer decay index [BDI]=2), and is diagnosed on the basis of eight synapomorphies, two of which are unique (marked here with *): anterior expansion of lachrymal (C.53); presence of multiple rami of infraorbital canal in jugal (C.56); mobile maxilla (C.73); presence of a peg-like anterior maxillary process (C.74*); presence of an interopercle (C.119*); internal carotid artery piercing the parasphenoid (C.180); presence of two ossifications of the ceratohyal (C.219); presence of ossified centra (C.287). These are all absent in †*Brachydegma*, which furthermore lacks any form of coronoid process, an essential component of the crown neopterygian hallmark. Therefore, †*Brachydegma* can be confidently excluded from the neopterygian crown, contrary to previous hypotheses (*Hurley et al., 2007*).

In our parsimony strict consensus tree (*Figure 11*), †*Brachydegma* is resolved as sister to a clade containing †birgeriids and †scanilepiforms + polypterids. This topology is weakly supported (BDI = 1) by three synapomorphies: extrascapular not reaching lateral margin of skull roof (C.45); presence of three or more suborbitals (C.55); differentiation of braincase ossifications (C.159). †*Birgeria* is resolved as sister to †scanilepiforms + polypterids, receiving equally low support (BDI = 1), on the basis of two ambiguous synapomorphies: imperforate hyomandibula (C.218); absence of a triradiate scapulocoracoid (C.243). We note ambiguities regarding this topology resulting from a poor understanding of the endoskeleton of †*Birgeria* (*Nielsen, 1949*).

Contrasting recent works (*Giles et al., 2017*; *Argyriou et al., 2018*; *Latimer and Giles, 2018*), †Saurichthyiformes are recovered as sister to Chondrostei (*Gardiner and Schaeffer, 1989*; *Gardiner et al., 2005*). Interestingly, a weakly supported (BDI = 2) deep-bodied clade of durophagous taxa – comprising †pycnodonts and †dapediids – is recovered within the neopterygian crown (contra *Latimer and Giles, 2018*) following the addition of †*Neoproscinetes penalvai*, the cranial endoskeleton of which (*Nursall and Maisey, 1987*; *Machado, 2008*) is better known than that of most other †pycnodonts (see also *Hurley et al., 2007*). This deep-bodied clade is resolved as sister to holosteans,

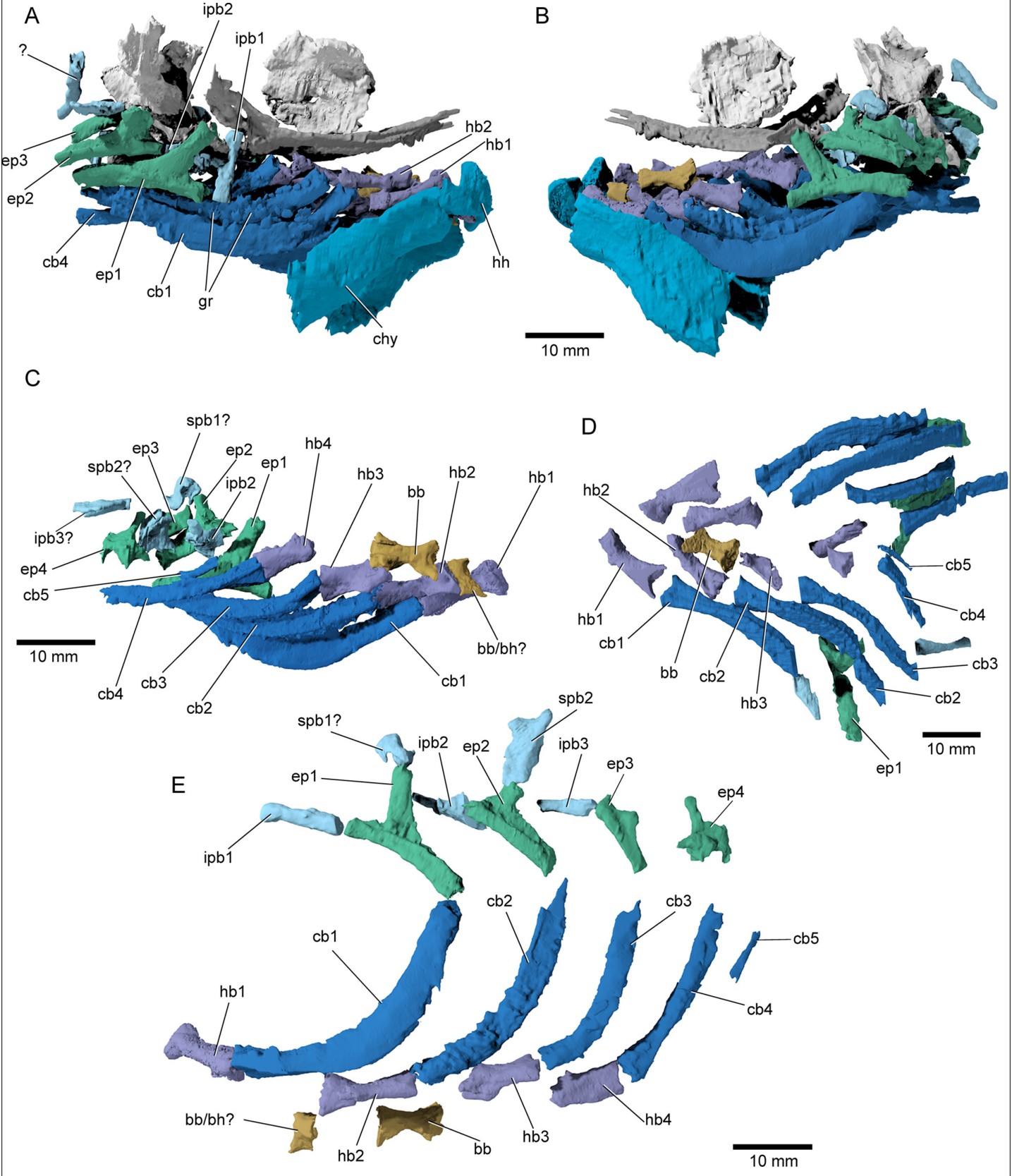

**Figure 9.** Branchial anatomy of †*Brachydegma caelatum*. Type specimen (MCZ VPF-6503): (**A**) right and (**B**) left lateral views of complete branchial and ventral hyoid skeleton shown as preserved; (**C**) branchial arches of left side in medial view; (**D**) ventral view of preserved branchial elements of paratype

*Figure 9 continued on next page*

(MCZ VPF-6504); (**E**) composite reconstruction of gill skeleton with branchial arches spaced and splayed ventrally. Abbreviations: **bb**, basibranchial; **bb/bh?**, basibranchial or basihyal; **cb1–5**, ceratobranchial 1–5; **chy**, ceratohyal; **ep1–4**, epibranchial 1–4; **gr**, gill rakers; **hb1–4**, hypobranchial 1–4; **hh**, hypohyal; **ipb1–3**, infrapharyngobranchial 1–3; **spb1,2**, suprapharyngobranchial 1,2. Scale bars equal 10 mm.

but the interrelationships of most of its constituents are unclear. These topologies are also present in the agreement subtrees (*Appendix 1—figure 3*).

*Constrained equally weighted parsimony analyses*: We ran additional unweighted parsimony experiments to investigate the number of steps needed to produce previously suggested topologies of †*Brachydegma*. When †*Brachydegma* is constrained with halecomorphs and †*Watsonulus* (*Hurley et al., 2007*), the resulting MPTs are 13 steps longer (1665 vs. 1652 steps). When †*Brachydegma* is constrained in a monophyletic clade with actinopterans (*Giles et al., 2017*), the analyses resulted in MPTs that are three steps longer (1655 steps) than the unconstrained parsimony analyses.

*Equally weighted parsimony excluding* †Brachydegma: When reanalyzing our phylogenetic matrix after excluding †*Brachydegma*, but keeping newly added/modified characters, clades recovered in the previous rounds of analyses are not resolved (1646 steps; *Appendix 1—figure 4*). Instead, the resolution of post-Devonian actinopterygian interrelationships is largely lost, and is replaced by a large polytomy containing major groups (e.g., †Scanilepiformes + Polypteridae, Chondrostei, and a poorly resolved neopterygian total group).

*Parsimony using implied weights*: A different phylogenetic picture of 'early' actinopterygian interrelationships emerges when analyzing our phylogenetic dataset with implied weights (*Goloboff, 1993*; *Goloboff et al., 2018*) using a gentle concave (K=12; see *Goloboff et al., 2018*). We used 44 best-fit trees (fit score = 68.77) to produce a strict consensus tree (*Appendix 1—figure 5*). In the strict consensus, †*Brachydegma* is resolved as a stem neopterygian, on the basis of three synapomorphies of varying fit scores (f): presence of a vertical preopercle (C.116; f:0.37); length of the median gular exceeding half the length of the lower jaw (C.124; f:0.14); and the presence of uncinate processes of the epibranchials (C.231; f:0.2). Still, †*Brachydegma* is clearly excluded from crown Neopterygii. Typically recognized neopterygian synapomorphies (see also equally weighted parsimony above) support a clade formed by †*Hulettia* (now excluded from crown Neopterygii) and crown neopterygians. These nine synapomorphies include: presence of multiple sensory rami on the jugal (C.56; f:0.45); mobile maxilla (C.73; f:0.2); presence of a peg-like anterior maxillary process (C.74*; f:0); presence of an angular and a surangular in the lower jaw (C.90; f:0.57); presence of an anterodorsal process on the subopercle (C.113; f:0.52); presence of an interopercle (C.120*; f:0.25); internal carotid artery piercing the parasphenoid (C.180; f:0.25); presence of two ossifications of the ceratohyal (C.219; f:0.2); presence of ossified centra (C.287; f:0.29). †*Brachydegma* shares with crown neopterygians + †*Hulettia* the same state for C.90, which is however widespread among actinopterygians. Other notable departures in tree shape from the unweighted and unconstrained analysis include: (i) the recovery of an 'Ancient fish clade' formed by Cladistia and Chondrostei and their fossil relatives; (ii) recovery of a clade formed by †Saurichthyiformes and †Birgeriidae at the base of the neopterygian stem.

*Bayesian analysis*: The exclusion of †*Brachydegma* from the neopterygian crown group is ratified in our Bayesian analysis (*Appendix 1—figure 6*), where the neopterygian crown is strongly supported (BPP=0.99). However, relationships outside of the neopterygian crown are volatile (compare, for example, *Gardiner and Schaeffer, 1989*; *Gardiner et al., 2005*; *Hurley et al., 2007*; *Giles et al., 2017*; *Argyriou et al., 2018*; *Latimer and Giles, 2018*; *Figueroa et al., 2019*) and poorly supported; for example, the actinopterygian crown node is not recovered. This picture does not change when †*Brachydegma* is removed from the matrix. We express caution in accepting hypotheses of relationships outside of the neopterygian crown.

## Discussion
### Comparative anatomy of †*Brachydegma*

†*Brachydegma* is thus far unique among described Permian actinopterygians in preserving the external dermal skeleton, braincase, hyoid arch, gill skeleton, and shoulder girdle in three dimensions.

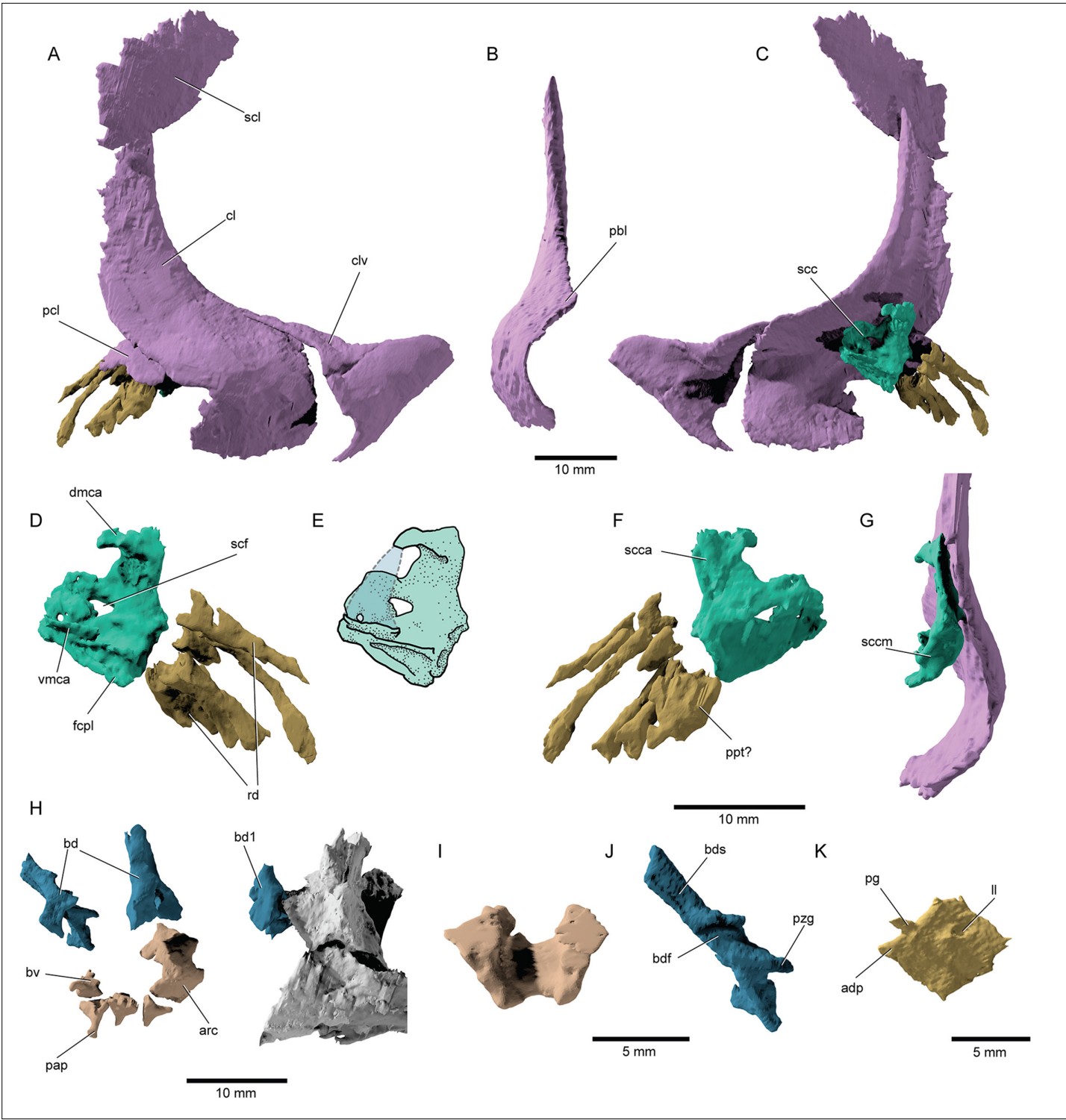

**Figure 10.** Pectoral fin and axial anatomy of †*Brachydegma caelatum*. Type specimen (MCZ VPF-6503): (**A**) Right pectoral girdle in lateral view; (**B**) anterior view of right cleithrum; (**C**) right pectoral girdle in medial view; (**D**) right scapulocoracoid and reconstructed fin ossifications in medial view; (**E**) line drawing of D with dotted line indicating the conceived course of the mesocoracoid arch; (**F**) right scapulocoracoid and reconstructed fin ossifications in medial view; (**G**) scapulocoracoid and cleithrum in posterior view; (**H**) anterior axial skeleton; (**I**) anterodorsal view of fused arcual element; (**J**) medial view of left basidorsal; (**K**) medial view of lateral line scale. Abbreviations: **adp**, anterodorsal process of scale; **bd**, basidorsal; **bdf**, medial furrow on basidorsal; **bds**, hemi-neural spine; **bv**, basiventral; **cl**, cleithrum; **clv**, clavicle; **dmca**, dorsal limit of the mesocoracoid arch; **fcpl**, facet for the coracoid plate; **ll**, lateral line pore; **pap**, parapophysis; **pbl**, post-branchial lamina; **pcl**, postcleithrum; **pg**, articular peg of scale; **ppt?**, putative propterygium; **pzg**, prezygapophysis; **rd**, radials; **scc**, scapulocoracoid; **sccm**, middle (articular) region of scapulocoracoid; **scf**, supracoracoid foramen;

*Figure 10 continued on next page*

*Figure 10 continued*

**scl**, supracleithrum; **vmca**, ventral limit of the mesocoracoid arch. Dashed lines represent hypothetical margins of mesocoracoid arch. Scale bars for A–H equal 10 mm; scale bars for I–K equal 5 mm.

It therefore represents an important addition between earlier Devonian-Carboniferous (e.g., †*Mimipiscis* [*Gardiner, 1984*]; †*Raynerius* [*Giles et al., 2015b*]; †*Lawrenciella* [*Poplin, 1974*; *Pradel et al., 2016*]; †*Trawdenia* [*Coates, 1999*; *Coates and Tietjen, 2018*]; †*Coccocephalichthys* [*Poplin and Véran, 1996*]) and later Triassic taxa (e.g., †*Watsonulus* [*Olsen, 1984*]; †*Saurichthys* [*Argyriou et al., 2018*]; †'*Perleidus*' and early teleosts [*Patterson, 1975*]) for which similar detailed anatomical information is available. †*Brachydegma* bears a novel combination of traits, but this is unsurprising given how little is known of the internal anatomy of any other Permian actinopterygian (but see *Gill, 1923*; *Aldinger, 1937*). The most notable new features of †*Brachydegma* revealed by our work relate to the braincase, palate, gill skeleton, and hyoid arch, as well as the pectoral endoskeleton and fin.

The braincase and parasphenoid of †*Brachydegma* display an unexpected combination of features. Rather than having a single, co-ossified occipital arch, †*Brachydegma* exhibits differentiated endochondral ossifications in the occipital region. This is the earliest example of a condition typically associated with neopterygians (*Patterson, 1975*; *Grande and Bemis, 1998*; *Grande, 2010*). The posterior parasphenoid stem extending behind the ventral otic fissure clearly differentiates †*Brachydegma* from most anatomically generalized Paleozoic-early Mesozoic actinopterygians (*Nielsen, 1942*; *Nielsen, 1949*; *Schaeffer and Dalquest, 1978*; *Gardiner, 1984*; *Giles et al., 2015b*; *Figueroa et al., 2019*). The bifurcation of the dorsal aorta into lateral dorsal aortae occurs below the posterior stem of the parasphenoid in †*Brachydegma*, resembling the condition seen in †saurichthyiforms (*Argyriou et al., 2018*), conceivably †birgeriids (*Nielsen, 1949*) and most actinopterans (*Patterson, 1975*), but not polypterids (*Allis, 1922*).

The presence of a dorsolateral metapterygoideal flange and groove in †*Brachydegma* is a possibly derived feature, which is encountered in a rudimentary form in †*Australosomus* (*Nielsen, 1949*), but is otherwise largely restricted to neopterygians (*Olsen, 1984*; *Arratia and Schultze, 1991*). In stem actinopterygians, like, for example, †*Mimipiscis*, or †*Pteronisculus*, the portion occupied by the metapterygoid – or its co-ossified homolog – does not bear a clearly defined lateral flange (*Nielsen, 1942*; *Gardiner, 1984*; *Arratia and Schultze, 1991*). By contrast to the above, the intimate contact between the palate and the maxilla, via overlapping flanges issuing from the ventral surfaces of both bones, is reminiscent of the primitive configuration seen in most stem and early diverging actinopterygians, and to an extent in *Polypterus* (*Nielsen, 1942*; *Nielsen, 1949*; *Gardiner, 1984*; *Giles et al., 2017*; *Argyriou et al., 2018*; *Lemberg et al., 2021*).

The nearly complete branchial skeleton of †*Brachydegma* represents one of the best examples known so far from a Paleozoic-early Mesozoic actinopterygian (*Stensiö, 1921*; *Stensiö, 1932*; *Aldinger, 1937*; *Nielsen, 1942*; *Nielsen, 1949*; *Giles et al., 2015b*; *Giles et al., 2017*; *Argyriou et al., 2018*; *Figueroa et al., 2019*). Unlike polypterids (*Allis, 1922*), †*Brachydegma* shares five gill arches – with the fifth represented only by a pair of tiny ceratobranchials – and four independent hypobranchial ossifications with most Permian-Triassic actinopterygians (*Nielsen, 1942*; *Nielsen, 1949*; *Giles et al., 2017*), chondrosteans (*Grande and Bemis, 1991*; *Hilton et al., 2011*), and most teleosts (*Nelson, 1969*; *Hilton, 2002*). The dorsal gill skeleton of †*Brachydegma* lacks enlarged tooth patches and has a well-ossified series of suprapharyngobranchials; both contrast with derived conditions found in neopterygians (*Nelson, 1969*; *Grande and Bemis, 1998*; *Hilton, 2002*; *Grande, 2010*).

Shoulder girdles and their patterns of variation remain poorly characterized in early fossil actinopterygians. The dermal shoulder girdle of †*Brachydegma* is well developed, comprising large supracleithra, presupracleithra, cleithra, postcleithra, and clavicles, differing from the reduced complement of dermal bones in neopterygians. However, the endoskeletal girdle is only partially mineralized, in contrast to those of most Devonian-Carboniferous taxa (*Gardiner, 1984*; *Coates and Tietjen, 2018*), but bearing some possible resemblance to the much-modified girdles of living actinopterygians (*Jessen, 1972*; *Grande and Bemis, 1998*; *Grande, 2010*; *Hilton et al., 2011*). †*Brachydegma* lacks a

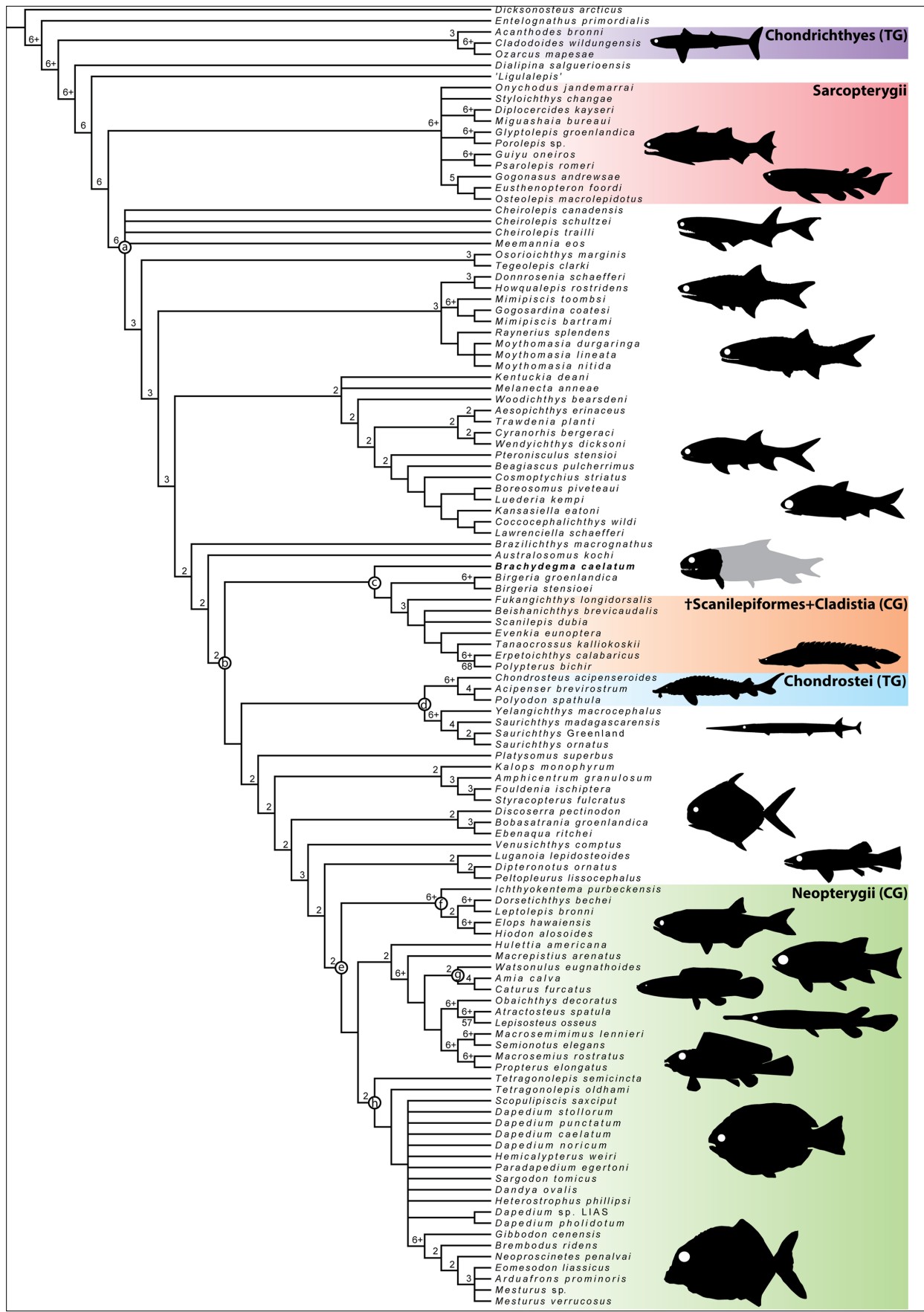

*Figure 11 continued on next page*

*Figure 11 continued*

**Figure 11.** Strict consensus of the 1412 most parsimonious trees of 1652 steps for 117 taxa and 300 equally weighted characters. Consistency index = 0.203, retention index = 0.66. Numbers above nodes indicate Bremer values above 1. Numbers below nodes indicate bootstrap percentages above 50%. Selected node optimizations are as follows: (**a**) (Actinopterygii total group): C.6 (1→0); C.29 (0→1,2); C.43 (0→1); C.45 (1→0); C.46 (0→1); C.54 (0→1); C.57 (0→1); C.63 (1→0); C.64 (0→1); C.69 (0→1); 70 (0→1); C.77 (0→1); C.79 (0→1); C.90 (0→1); C.109 (0→1); C.139 (0→1); C.152 (0→1); C.199 (0→1); C.215 (0→1); C.257 (0→1); C.258 (0→1); C.264 (0→1); (**b**) (Actinopterygii crown group): C.67 (1→0); C.101 (0→1); C.103 (0→1); C.107 (0→1); C.157 (0→2); C.174 (0→2); (**c**) (†*Brachydegma* + (†Birgeriidae+(†Scanilepiforms + †Polypteridae))): C.45 (1→0); C.55 (0→3); C.158 (0→1); (**d**) (Chondrostei + †Saurichthyiformes): C.14 (1→0); C.111 (1→2); C.123 (1→0); C.210 (1→0); (**e**) (Neopterygii crown group): C.53 (0→1); C.56 (0→1); C.73 (0→1); C.74 (0→1); C.119 (0→1); C.180 (0→1); C.219 (0→1); C.287 (0→1); (**f**) (Teleostei total group): C.9 (0→1); C.47 (1→0); C.55 (3→1); C.168 (0→1); C.169 (0→1); C.225 (0→1); C.270 (0→1); (**g**) (Halecomorphi + †*Watsonulus*): C.75 (0→1); C.76 (0→1); C.97 (0→1); C.135 (0→1); C.220 (1→0); C.280 (1→0); (**h**) ((†Dapediidae +†Pycnodontiformes) + †*Tetragonolepis*): C.59 (2→1); C.223 (0→1); C.255 (0→2); C.266 (0→1); C.278 (0→1); C.284 (0→1); C.285 (0→1); C.289 (0→1).

mineralized coracoid plate or completely mineralized mesocoracoid arch, in contrast to most deeply diverging actinopterygians (*Nielsen, 1942*; *Jessen, 1972*; *Gardiner, 1984*; *Coates and Tietjen, 2018*) and the Triassic neopterygian †*Watsonulus* (*Olsen, 1984*). We also found no evidence of a posterior mesocoracoid process, as that present in, for example, †*Mimipiscis* (*Gardiner, 1984*) or †*Trawdenia* (*Coates and Tietjen, 2018*). The lack of a dorsal scapular process resembles stem actinopterygians such as †*Trawdenia* (*Coates and Tietjen, 2018*), and clearly differs from the earliest neopterygian pectoral girdle known, that of †*Watsonulus* (*Olsen, 1984*). However, the endoskeleton of living actinopterygian taxa presents conflicting anatomical data, and the evolution of these features is difficult to resolve without additional information from fossils.

Perhaps most surprisingly, the hyoid arch of †*Brachydegma* shows the presence of two accessory hyoid elements between the dorsal and ventral components of the arch. These two ossifications exhibit a sub-parallel arrangement, with the anteriormost articulating with the lower jaw to form a double jaw joint.

## Phylogenetic position of †*Brachydegma*

The unanticipated combination of characters found in †*Brachydegma* is associated with ambiguity in its phylogenetic position, with equal weights parsimony, implied weights parsimony, and Bayesian analyses yielding conflicting placements. Although equal weights parsimony analyses suggested a close affinity with polypterids, †scanilepiforms and †*Birgeria* (*Figure 11*) implied weights parsimony recovered †*Brachydegma* as a stem neopterygian (*Appendix 1—figure 5*). Constrained analyses emulating previously proposed topologies for †*Brachydegma* as a stem actinopteran (*Giles et al., 2017*) or a halecomorph (*Hurley et al., 2007*) also resulted in longer trees. The volatility of early actinopterygian phylogeny between successive studies using similar character sets but different taxon samples suggests that these hypotheses should be viewed with caution, until more information of both the internal and external anatomy of additional late Paleozoic-early Mesozoic fossil groups becomes available (*Giles et al., 2017*; *Argyriou et al., 2018*; *Latimer and Giles, 2018*).

Our reappraisal of the systematic affinities of †*Brachydegma* contrasts sharply with past hypotheses of crown neopterygian affinity (*Hurley et al., 2007*). Evidence previously advanced for a neopterygian (and specifically halecomorph) placement of †*Brachydegma* included the presence of: a supraorbital bone; a large median gular; a posteriorly indented and possibly free maxilla; an antorbital with a tube-like anterior process; and a neopterygian-like coronoid process (*Hurley et al., 2007*). The majority of these characters are either widely distributed across actinopterygians (supraorbitals, antorbital with a tube-like anterior process) or were misidentified in †*Brachydegma* (posteriorly indented maxilla, free maxilla, coronoid process, and the possible misidentification of the nasal as an antorbital). Our analyses recover †*Brachydegma* firmly outside the neopterygian crown (see also constrained analyses). This is despite the presence of features of the hyoid arch formerly thought to be restricted to crown neopterygians – halecomorphs in particular, the implications of which we discuss in greater detail below.

## Evolution of accessory elements in the hyoid arch of actinopterygians

The homology and evolution of hyoid elements is complex, and our new data on the hyoid arch of †*Brachydegma* add to this long-running debate (*Patterson, 1973*; *Jollie, 1980*; *Patterson, 1982*; *Jollie, 1984*; *Véran, 1988*; *Gardiner and Schaeffer, 1989*; *Gardiner et al., 2005*; *Warth et al., 2017*).

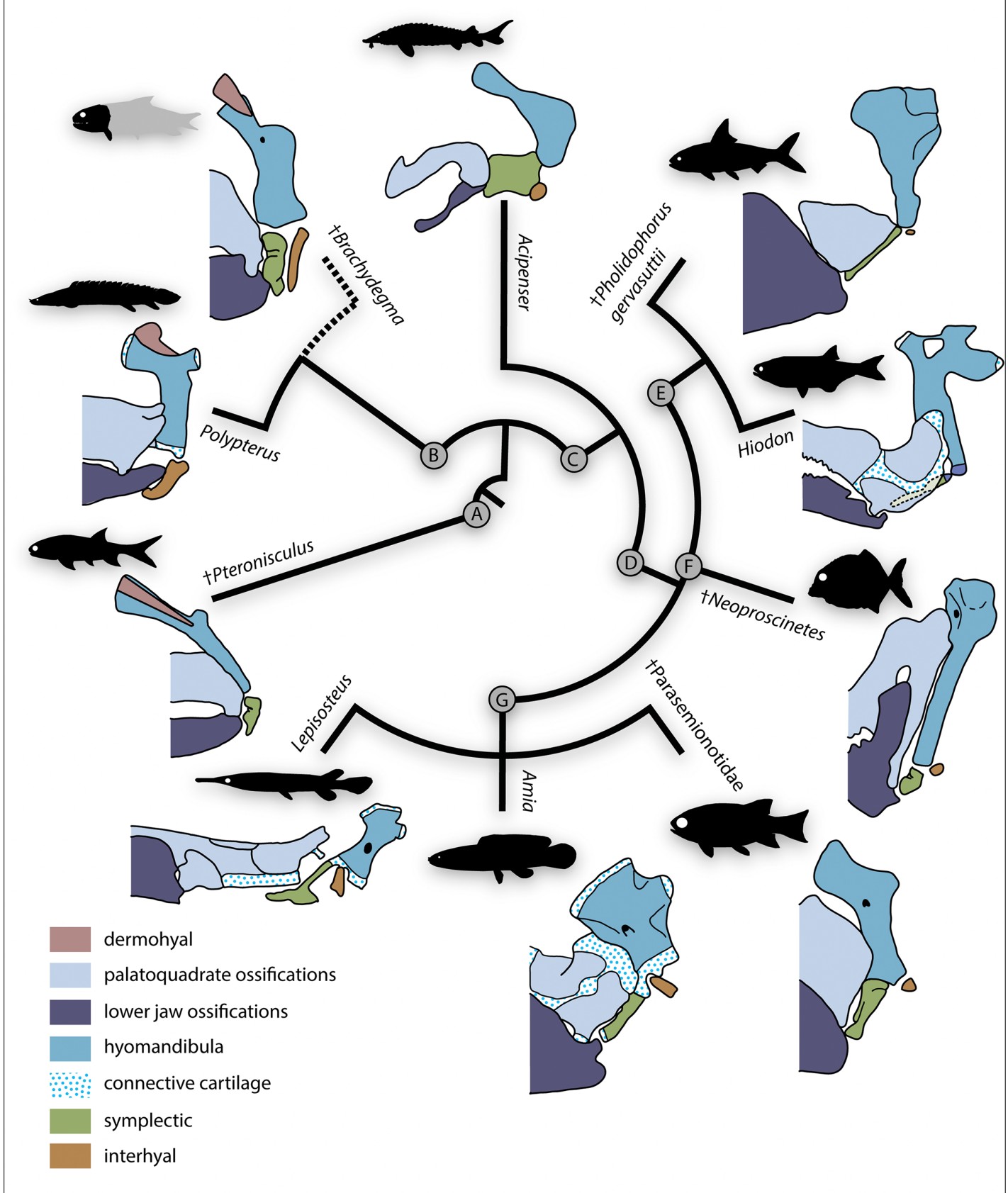

**Figure 12.** Evolutionary morphology of accessory hyoidean elements of actinopterygians. Simplified evolutionary hypothesis from *Figure 11*. Nodes are as follows: (**A**) stem Actinopterygii; (**B**) †*Brachydegma* + Polypteridae; (**C**) Actinopteri; (**D**) Neopterygii; (**E**) Teleostei; (**F**) †Pycnodontiformes; (**G**) Holostei. *Figure 12 continued on next page*

*Figure 12 continued*

Sources of anatomical information: †*Pteronisculus*, †*Brachydegma*, *Acipenser*, †Parasemionotidae indet, this work; *Polypterus* (**Allis, 1922**; **Jollie, 1984**); †*Pholidophorus gervasuttii* (**Arratia, 2013**); *Hiodon*, with cartilaginous interhyal omitted as it does not articulate with the hyomandibula (**Hilton, 2002**); †*Neoproscinetes* (**Nursall and Maisey, 1987**; **Gardiner et al., 1996**); *Lepisosteus* (**Grande, 2010**); *Amia* (**Grande and Bemis, 1998**). Drawings not to scale.

Here, we review past interpretations of accessory hyoid elements in living and fossil actinopterygians, and attempt to synthesize these data in light of new information from †*Brachydegma* (*Figure 12*).

Polypterids possess a single accessory hyoid element that articulates with the full width of the hyomandibula dorsally and of the ceratohyal ventrally (**Allis, 1922**; **Moy-thomas, 1933**; **Jollie, 1984**; **Giles et al., 2017**). This bone develops independently of the hyosymplectic cartilage and thus represents an interhyal (**Moy-thomas, 1933**; **Jollie, 1984**). Neopterygians primitively have two accessory elements between the hyomandibula and ceratohyal. The more anterior of these – the symplectic – typically braces the mandibular arch (**Patterson, 1973**; **Olsen, 1984**; **Véran, 1988**; **Gardiner and Schaeffer, 1989**; **Gardiner et al., 1996**; **Grande and Bemis, 1998**; **Grande, 2010**; **Arratia, 2013**) and arises as an anteroventral outgrowth of the embryonic hyosymplectic cartilage that subsequently detaches in development (**Holmgren, 1943**; **Bertmar, 1959**; **Konstantinidis et al., 2015**; **Mork and Crump, 2015**; **DeLaurier, 2019**). The more posterior accessory element – the interhyal – articulates with the posteroventral portion of the hyomandibula and suspends the ventral hyoid elements (**Patterson, 1982**; **Véran, 1988**; **Grande and Bemis, 1998**; **Grande, 2010**; **Konstantinidis et al., 2015**; **DeLaurier, 2019**). The interhyal arises from an independent embryonic cartilage (**Konstantinidis et al., 2015**; **Mork and Crump, 2015**; **DeLaurier, 2019**). The interhyal remains cartilaginous in adult holosteans (**Grande and Bemis, 1998**; **Grande, 2010**), while the symplectic shows contrasting conditions in different members of that group. In halecomorphs, it is hatchet shaped and articulates with the articular forming a second jaw joint (**Grande and Bemis, 1998**). In ginglymodans, it is 'L' shaped and joins the back of the primary palate (**Grande, 2010**). In extant teleosts, the symplectic is wedge-shaped and typically inserts in a notch on the quadrate, while the interhyal is variably ossified and lies between the hyomandibula and posterior ceratohyal (**Hilton, 2002**; **Arratia, 2013**).

In contrast to the uncontroversial assessments for polypterids and neopterygians, both the number and identity of accessory hyoid elements in acipenseriforms has been the subject of considerable debate. Acipenseriform hyoid and mandibular arches are highly modified with respect to those of other actinopterygians. The anteroventral tip of the hyomandibula in acipenseriforms articulates with a large cartilage or bone that in turn is linked with the palatoquadrate. This first bone or cartilage articulates ventrally with a second, often much smaller one, which articulates with a bone universally regarded as a ceratohyal. Two principal interpretations have been offered for the elements between the ceratohyal and hyomandibula. The first, initially proposed by *Traquair, 1877* and adopted by several subsequent authors (**Sewertzoff, 1928**; **Marinelli and Strenger, 1973**; **Jollie, 1980**; **Véran, 1988**), is that there are two accessory hyoid elements: the dorsal one representing a symplectic, and the ventral one representing an interhyal. The second, most forcefully argued by *Patterson, 1982*, posits that the dorsal bone or cartilage is an interhyal. Under this interpretation, the smaller ventral cartilage is not an accessory element but rather a posterior ceratohyal, as indicated by a close association with a branchiostegal ray in *Polyodon* (**Patterson, 1982**). These competing models have contrasting implications for character distribution in actinopterygians: the first suggests that a symplectic is a feature of actinopterans rather than neopterygians, while the second preserves the status of the symplectic as a neopterygian neomorph. Patterson's (1982) model is now dominant (e.g., **Gardiner, 1984**; **Gardiner and Schaeffer, 1989**; **Gardiner et al., 1996**; **Coates, 1999**; **Gardiner et al., 2005**; **Grande, 2010**; **Xu et al., 2014**; **Warth et al., 2017**; **López-Arbarello and Sferco, 2018**; **Xu, 2019**), but neither scheme is without its challenges. The interpretation of these elements as a symplectic and interhyal, respectively, requires a change in developmental pattern between the neopterygian (derived from common hyosymplectic cartilage: **Konstantinidis et al., 2015**; **Mork and Crump, 2015**) and chondrostean symplectic (a separate cartilage throughout development: **Warth et al., 2017**) plus the neomorphic – but not unprecedented (see examples in teleosts: **McAllister, 1968**) – association between a branchiostegal and interhyal in paddlefishes (**Patterson, 1982**; **Grande and Bemis, 1991**). The alternative interpretation requires the origin of a symplectic-like association between the interhyal and palatoquadrate, and the migration of the ceratomandibular ligament from

the posterior ceratohyal to the anterior, combined with the development of the posterior ceratohyal matching that of an interhyal (derived from an independent cartilage) rather than a neopterygian posterior ceratohyal (ossifies within the same cartilage as the anterior ceratohyal; *Warth et al., 2017*).

It is within this limited framework that previous authors have tried to interpret the accessory hyoid ossifications of fossil actinopterygians. †Parasemionotids (*Stensiö, 1932*; *Patterson, 1973*; *Olsen, 1984*, *Figure 8F–I*) and †pycnodonts (*Nursall and Maisey, 1987*; *Gardiner et al., 1996*; *Kriwet, 2005*) have two accessory ossifications, which are unambiguously identified as an interhyal and symplectic. In both groups, these two elements are arranged in a sub-parallel manner and the symplectic articulates with the lower jaw forming a double jaw joint. *Arratia, 2013*, argues that some stem teleosts (e.g., †*Pholidophorus gervasuttii*) likewise have a double jaw joint; the broad distribution of this character raises questions as to its reliability as a halecomorph synapomorphy. Interpretation of accessory elements in groups more remote from the neopterygian crown is less straightforward. This reflects a series of obstacles related to both fossils (distortion of spatial relationships, an inability to distinguish genuine absence of a structure from its failure to ossify, uncertain phylogenetic placements) and living taxa (unsettled interpretations of homologies). Most probable stem actinopterygians exhibit (or, more properly, preserve) a single ossified accessory hyoid element (*Nielsen, 1942*; *Nielsen, 1949*; *Gardiner, 1984*, *Figure 8J–M*), conventionally identified as the interhyal following *Patterson, 1982*. However, *Véran, 1988*, reported a second ossification in some fossils, found in close association with the anteroventral tip of the hyomandibula and the palatoquadrate. *Véran, 1988*, argued this second bone is a symplectic, and that both this and an interhyal were primitively present in actinopterygians (see also *Olsen, 1984*). In some probable stem actinopterygians of Triassic-Jurassic age (e.g., †*Boreosomus reuterskioldi*, †'*Pteronisculus*' *gyrolepidoides*, †*Ptycholepis bollensis*), these two intermediate elements appear to be arranged in a dorsoventral series, rather than in a sub-parallel manner (*Véran, 1988*). This hypothesis gained little traction, however, and subsequent authors reinterpreted putative symplectics in extinct non-neopterygians as interhyals, with the bones reported by Véran as interhyals reclassified as posterior ceratohyals or articulars (*Gardiner and Schaeffer, 1989*; *Patterson, 1994*; *Gardiner et al., 1996*; *Gardiner et al., 2005*). A key argument for the dismissal of putative non-neopterygian symplectics was the apparent absence of a break in the perichondral lining of the so-called symplectic, which would imply the articulation of the symplectic with the lower jaw in a neopterygian manner (*Patterson, 1994*; *Gardiner et al., 1996*).

†*Brachydegma* provides a new perspective on this debate (*Figures 8 and 12*) for two reasons. First, it preserves two accessory hyoid elements in three dimensions and life position. Second, its phylogenetic position, although ambiguous, clearly lies far outside halecomorphs, or the neopterygian crown. In this context, it is significant that the more anterior element matches structural criteria used to identify the symplectic in crown neopterygians: it lies immediately posterior to the quadrate, between the hyomandibula and lower jaw, and it forms a clear articulation with the latter via a condyle (*Patterson, 1973*; *Patterson, 1982*; *Gardiner et al., 1996*; *Grande and Bemis, 1998*). A further similarity between this anterior element of †*Brachydegma* and the undisputed symplectic of †parasemionotids is the presence of an aortic groove (sensu *Véran, 1988*) on the dorsal anterolateral and lateral surface of the bone (*Figure 8*). Under this interpretation, the slender, rod-shaped posterior element of †*Brachydegma* would represent an interhyal.

Our µCT-aided examination of the hyoid arch of the putative stem actinopterygian †*Pteronisculus gunnari* revealed a single accessory element (*Figure 8J–M*), which presents many similarities with the symplectic of †*Brachydegma* and also satisfies most criteria for establishing its homology with the neopterygian symplectic. Specifically, it: (i) forms an anterior thickening – but not a condyle – for attachment to the lower jaw; (ii) bears an arterial groove (sensu *Véran, 1988*); and (iii) displays a topology identical to that of the neopterygian symplectic, as well as the now-recognized symplectic of †*Brachydegma*. We note that there is no concave socket in the articular of †*Pteronisculus* for the insertion of the anterior thickening of the 'symplectic', but rather a flat surface. Similar features and geometries to those of †*Pteronisculus* were also recognized in the 'symplectics' of putative stem actinopterygians, such as †*Boreosomus*, †*Ptycholepis*, †*Acrorhabdus,* and †*Pteronisculus* (*Véran, 1988*). These similarities provide support to previous hypotheses for a widespread distribution of the presence of a symplectic in actinopterygian fishes (*Olsen, 1984*; *Véran, 1988*). Finally, direct observations (TA) on mechanically prepared specimens of †*Boreosomus reuterskioldi* (MNHN.F SVT 14a; MNHN.F SVT 15b; *Appendix 1—figure 7E and F*), previously studied by *Véran, 1988*, confirmed the presence of

two accessory hyoid elements: a symplectic-like bone and an interhyal. Both accessory elements lie in series, as depicted by Véran (1998:pl. 1; *Appendix 1—figure 7E and F*).

Direct homology between the arrangement in †*Brachydegma*, crown neopterygians and stem actinopterygians, would also bolster arguments that the large accessory hyoidean cartilage of chondrosteans is a symplectic (*Figure 8N–Q*; *Figure 12*). Establishing these homologies faces outstanding challenges, including the scarcity of reliable accessory hyoid data for extinct actinopterygians, lack of developmental information in fossils, and the difficulty in identifying true absence from taphonomic loss or persistence as a cartilage. Only the first of these issues can be addressed, the path toward its resolution is clear: systematic re-examination of hyoid-arch structure in fossil taxa. The resolution of key issues concerning hyoid arch evolution in actinopterygians – including the homology of accessory elements across living lineages – will only be possible when high-quality anatomical data are available for a range of fossil taxa with well-supported phylogenetic placements. Despite uncertainties regarding the precise evolutionary affinities of †*Brachydegma* with regards to crown actinopterygian groups, the discovery of a double jaw joint in the former taxon suggests, at minimum, that this feature can no longer be treated as unique synapomorphy for halecomorphs. The possibility of a symplectic, or a double jaw joint evolving convergently (*Xu, 2019*) in halecomorphs and other fossil groups, including †*Brachydegma,* †pycnodonts, and possibly early teleosts, becomes very remote when accounting for the likely presence of these features in stem actinopterygians.

## Materials and methods

### Institutional abbreviations

MCZ: Museum of Comparative Zoology, Harvard University, Cambridge, MA, USA; MNHN: Muséum National d'Histoire Naturelle, Paris, France; NHMD: Natural History Museum of Denmark, University of Copenhagen, Copenhagen, Denmark; UMMZ: University of Michigan Museum of Zoology, Ann Arbor, MI, USA.

### Comparative material

NHMD 73588A, †*Pteronisculus gunnari*, holotype preserving cranial skeleton, Early Triassic, East Greenland; NHMD 74424A (*Appendix 1—figure 7A, B*), †Parasemionotidae indet. Early Triassic, East Greenland; MNHN.F SVT 14a and MNHN.F SVT 15b, †*Boreosomus reuterskioldi*, mechanically prepared crania with mandibular, hyoidean, and gill arches preserved in situ, Early Triassic, Spitsbergen (*Appendix 1—figure 7C, D*). The latter two specimens were studied by *Véran, 1988*; UMMZ 64250, *Acipenser brevirostrum*, scan of PTA-stained head.

### X-ray computed microtomography

μCT of the two specimens of †*Brachydegma* was performed with a Nikon XT H 225 ST scanner at the CTEES lab in the Department of Earth and Environmental Sciences, University of Michigan. The parameters are as follows: MCZ VPF-6503: 200 kV, 200 μA, 1.25 mm copper filter, giving a voxel size of 48.4 μm; MCZ VPF-6504: 215 kV, 265 μA, 3.5 mm copper filter, giving a voxel size of 61.2 μm. The head of *Acipenser brevirostrum* (UMMZ 64250) was also scanned using the same facilities, and the parameters are: 75 kV, 290 μA, no filtering, giving a voxel size of 60.2 μm. μCTs of †*Pteronisculus gunnari* (NHMD 73588A) and the †parasemionotid (NHMD 74424A) were performed at the University of Bristol using a Nikon XT H 225 ST scanner. The parameters are as follows: NHMD 73588: 224 kV, 191 μA, 1 mm copper filter giving a voxel size of 22.5 μm; NHMD 74424a: 223 kV, 155 μA, 0.5 mm copper filter, giving a voxel size of 20.3 μm. The resulting tomograms were processed in Mimics (biomedical.materialise.com/mimics; Materialise, Leuven, Belgium) for the creation of three-dimensional, digital anatomical models. The reconstruction process of the two †*Brachydegma* specimens was challenging, since the accommodating matrix is particularly rich in radiodense content. In the case of MCZ VPF-6503, the external surfaces of endoskeletal elements are lined with a dense mineral layer, hampering beam penetration and voxel size of smaller structures, such as nerve foramina. However, we were able to reconstruct the gross morphology of the endoskeleton. The completed models were exported in .ply format, and processed in Blender (https://www.blender.org/) for imaging.

## Phylogenetic analysis

For analyzing the interrelationships of †*Brachydegma* in a broader osteichthyan context, we modified an already existing, large-scale phylogenetic matrix (*Giles et al., 2017*; *Argyriou et al., 2018*; *Latimer and Giles, 2018*; *Figueroa et al., 2019*; see Appendix 2 for details). The matrix was edited in Mesquite (*Maddison and Maddison, 2017*), and the parsimony analyses were performed with 'New Technology Search' implemented in TNT (*Goloboff et al., 2008*; *Goloboff and Catalano, 2016*). We used the windows-based version of TNT. We enforced an outgroup constraint to ensure the monophyly of Osteichthyes. Relative fit difference was set to 0.1 and suboptimal trees up to 10 steps longer were retained. Initial trees were created by 1000 random addition sequences using 100 iterations or rounds of the four 'New Technology search' algorithms (Sectorial Search, Ratchet, Drift, Tree Fusing). For Sectorial Search algorithms, minimum sector size was set to five and maximum sector size was set to 58, which corresponds to ~50% taxa in our matrix. All other parameters remained unchanged. To ensure an exhaustive search of the dataset, two separate analyses of three rounds each were conducted using an alternation of 1000 iterations of 'Ratchet' and 'Sectorial Search' algorithms. The first analysis comprised a round of Sectorial Search, followed by two rounds using Ratchet. The second analysis started with a Ratchet round, followed by a round of Sectorial Search and then another round of Ratchet. All trees including suboptimals were saved at the end of each round, but only optimal trees (MPTs) were kept in memory for running the following round. Each round in both analyses was always complemented by the 1000 iterations of 'Tree Fusing'. Suboptimal trees from all rounds of analyses were used at the end to calculate Bremer supports (BDI) in TNT. Bootstrap values were calculated by reanalyzing the matrix with 10,000 iterations of the 'Traditional Search' algorithm. Agreement subtrees were also produced using the relevant function in TNT. Consistency (CI) and Retention (RI) indices were calculated using Mesquite. Additional analyses following the same methodology were run with constraints to investigate previous hypotheses for the placement of †*Brachydegma*. The same analysis procedure was replicated for producing the implied weights analyses, after selecting the relevant function in TNT and setting the constant K to K=12 (*Goloboff et al., 2018*). The latter analysis was conducted in order to visualize the interrelationships of actinopterygians – †*Brachydegma* in particular – in a scenario that softly penalizes homoplasy (homoplasic character downweighting), which is otherwise widespread in our unweighted parsimony generated trees. Characters are assigned fit values (f) between 0 and 1, with a value of 0 indicating that the character is not homoplastic (see, for example, C.74).

We also conducted a Bayesian analysis of our dataset in MrBayes (*Ronquist et al., 2012*), employing the same outgroup constraint as applied in parsimony analyses. The datatype was set to 'standard' (=morphological). We specified a gamma distribution for rates of character evolution, and indicated that invariant characters were not included in the matrix. We conducted two runs using the default of four chains, one cold and three heated. We assessed convergence by examining: the standard deviations of split frequencies; ESS values; and visual inspection of the trace of log likelihoods. We discarded the first 50% of sampled trees as burnin.

## Acknowledgements

S Pierce and J Cundiff (both MCZ), D Nelson (UMMZ), BEK Lindow and KM Gregersen (both NHMD) are thanked for kindly providing access to fossil and/or recent material. MR Sánchez-Villagra, C Romano (both PIMUZ), M Coates (University of Chicago), T Miyashita (Canadian Museum of Nature), T Simões (Harvard University), O Vernygora (University of Kentucky), A López-Arbarello (LMU), and M Véran (MNHN) are thanked for useful discussions. MR Sánchez-Villagra is also thanked for additional administrative support to TA. A Pradel and G Clément (both MNHN) are thanked for facilitating access to the collections of the MNHN. D Germain and P Vincent (both MNHN) and the FOSFO team of the CR2P are acknowledged for further facilitating the research activities of TA at the MNHN. C Abraczinskas (UMMP) created the weighted-line drawings of the holotype (Figures 2–4). Christina Byrd (MCZ) provided guidance in archiving data in Morphosource. We thank the reviewing editor, Zhu Min (Institute of Vertebrate Paleontology and Paleoanthropology, Chinese Academy of Sciences), P Ahlberg (Uppsala University), and an anonymous referee for their constructive feedback that improved earlier versions of this contribution. The Willi Hennig Society is acknowledged for making TNT available free of charge. This study includes data produced in the CTEES facility at University

of Michigan, supported by the Department of Earth and Environmental Sciences and the College of Literature, Science, and the Arts. Funding: This research was supported by P1ZHP3_168253 and P2ZHP3_184216 Swiss National Science Foundation grants to TA. Support during the late stages of this work was provided by an Alexander von Humboldt fellowship to TA. SG was supported by Royal Society Dorothy Hodgkin Research Fellowship (DH160098).

## Additional information

### Funding

| Funder | Grant reference number | Author |
|---|---|---|
| Schweizerischer Nationalfonds zur Förderung der Wissenschaftlichen Forschung | P1ZHP3_168253 | Thodoris Argyriou |
| Schweizerischer Nationalfonds zur Förderung der Wissenschaftlichen Forschung | P2ZHP3_184216 | Thodoris Argyriou |
| Alexander von Humboldt-Stiftung | | Thodoris Argyriou |
| Royal Society | Dorothy Hodgkin Research Fellowship (DH160098) | Sam Giles |

The funders had no role in study design, data collection and interpretation, or the decision to submit the work for publication.

### Author contributions

Thodoris Argyriou, Conceptualization, Data curation, Formal analysis, Funding acquisition, Investigation, Methodology, Project administration, Visualization, Writing – original draft; Sam Giles, Formal analysis, Investigation, Methodology, Validation, Visualization, Writing - review and editing; Matt Friedman, Conceptualization, Funding acquisition, Investigation, Methodology, Project administration, Resources, Supervision, Validation, Writing - review and editing

### Author ORCIDs

Thodoris Argyriou http://orcid.org/0000-0002-2036-5088
Sam Giles http://orcid.org/0000-0001-9267-4392
Matt Friedman http://orcid.org/0000-0002-0114-7384

### Decision letter and Author response

Decision letter https://doi.org/10.7554/eLife.58433.sa1
Author response https://doi.org/10.7554/eLife.58433.sa2

## Additional files

### Supplementary files

• Transparent reporting form

### Data availability

μCT raw and/or derived data are available on Morphosource. Links to parent directories for each studied specimen are given below. †*Brachydegma caelatum* (MCZ VPF-6503): http://www.morphosource.org/concern/media/000440974 †*Brachydegma caelatum* (MCZ VPF-6504): http://www.morphosource.org/concern/media/000441020 †*Pteronisculus gunnari* (NHMD VP 73588A): http://www.morphosource.org/concern/media/000441157 †Parasemionotidae indet. (NHMD VP 74424A): http://www.morphosource.org/concern/media/000441197 *Acipenser brevirostrum* (UMMZ 64250):

*Continued*

http://www.morphosource.org/concern/media/000441184 Phylogenetic matrix and trees available through Dryad at: https://doi.org/10.5061/dryad.jsxksn0bz.

The following datasets were generated:

| Author(s) | Year | Dataset title | Dataset URL | Database and Identifier |
|---|---|---|---|---|
| Argyriou T, Giles S, Friedman M | 2022 | A Permian fish reveals widespread distribution of neopterygian-like jaw suspension | https://doi.org/10.5061/dryad.jsxksn0bz | Dryad Digital Repository, 10.5061/dryad.jsxksn0bz |
| Argyriou T, Giles S, Friedman M | 2022 | Data (Raw scan data and derived data and digital renders for MCZ_VPF_6503) from: A Permian fish reveals widespread distribution of neopterygian-like jaw suspension | http://www.morphosource.org/concern/media/000440974 | Morphosource, 000440974 |
| Argyriou T, Giles S, Friedman M | 2022 | Data (Raw scan data and derived data and digital renders for MCZ_VPF_6504) from: A Permian fish reveals widespread distribution of neopterygian-like jaw suspension | http://www.morphosource.org/concern/media/000441020 | Morphosource, 000441020 |
| Argyriou T, Giles S, Friedman M | 2022 | Data (digital anatomical renders for NHMD VP 73588A) from: A Permian fish reveals widespread distribution of neopterygian-like jaw suspension | http://www.morphosource.org/concern/media/000441157 | Morphosource, 000441157 |
| Argyriou T, Giles S, Friedman M | 2022 | Data (digital anatomical renders for NHMD VP 74424A) from: A Permian fish reveals widespread distribution of neopterygian-like jaw suspension | http://www.morphosource.org/concern/media/000441197 | Morphosource, 000441197 |
| Argyriou T, Giles S, Friedman M | 2022 | Data (digital anatomical renders for UMMZ 64250) from: A Permian fish reveals widespread distribution of neopterygian-like jaw suspension | http://www.morphosource.org/concern/media/000441184 | Morphosource, 000441184 |

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

## Appendix 1

### Additional figures and phylogenetic trees

A

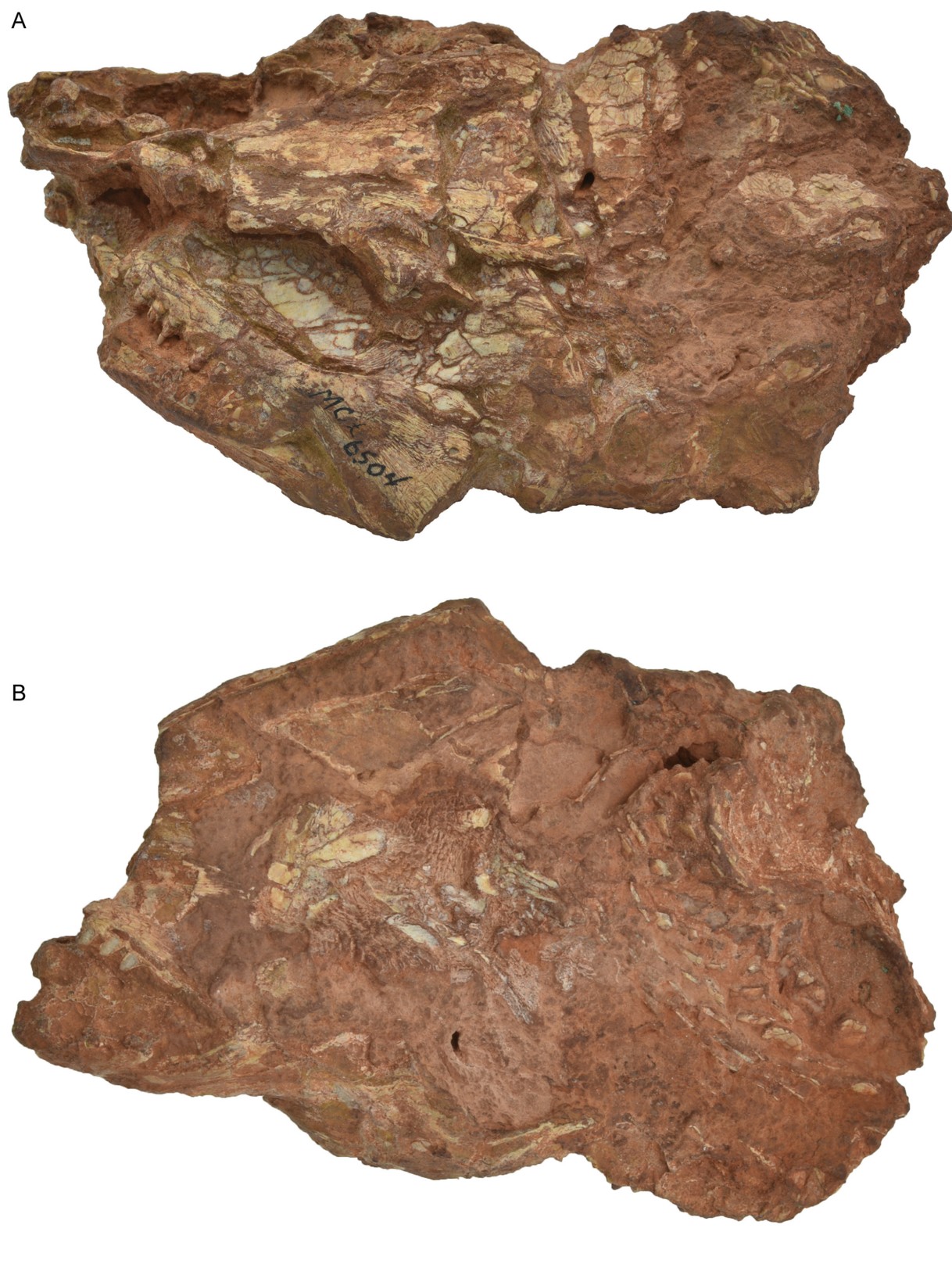

B

10 mm

**Appendix 1—figure 1.** †*Brachydegma caelatum* paratype specimen (MCZ VPF-6504). Compressed specimen in (**A**) dorsolateral and (**B**) ventrolateral (bottom) views.

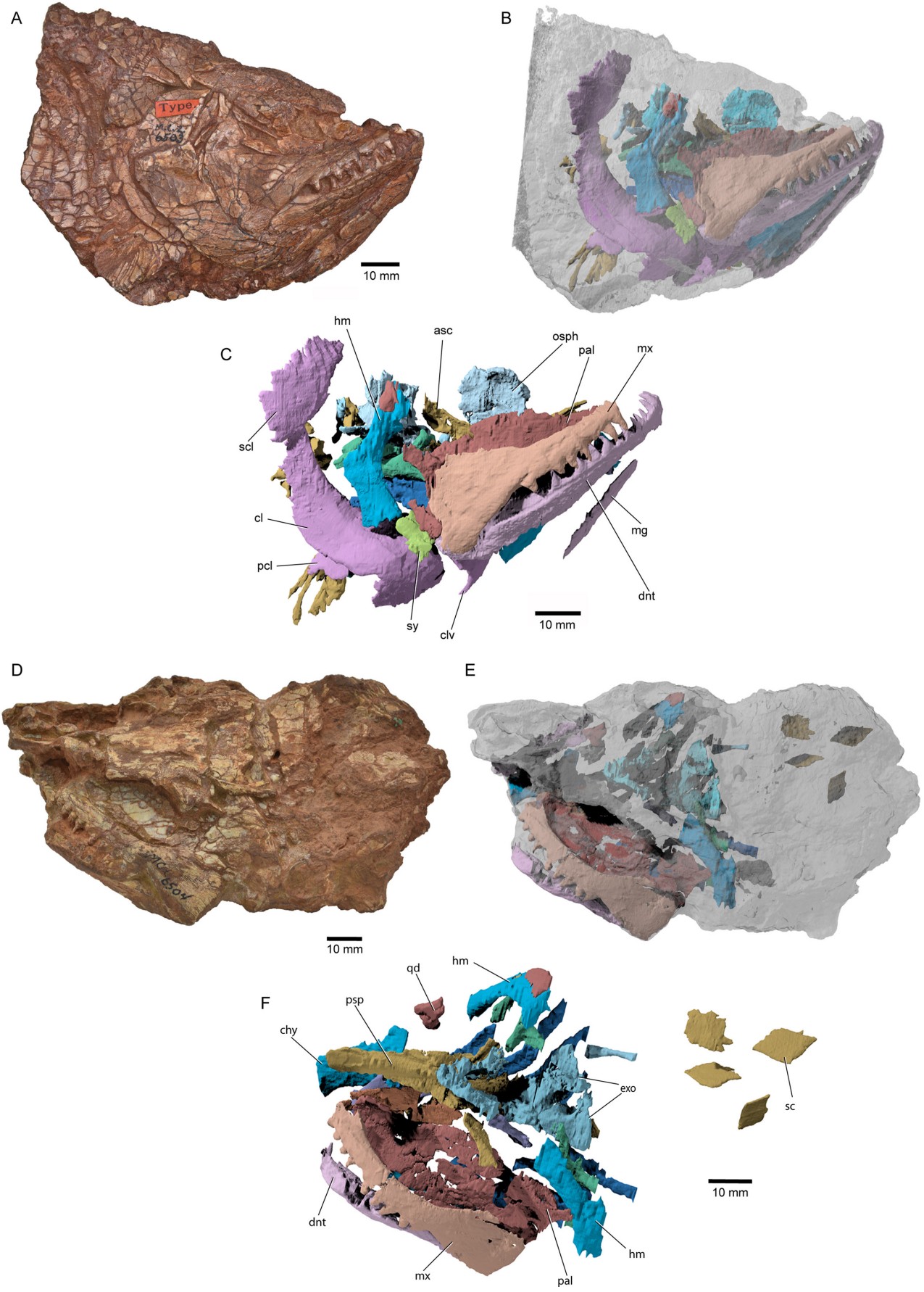

**Appendix 1—figure 2.** 3D-reconstructed portions of †*Brachydegma caelatum*. Type specimen (MCZ VPF-6503): (**A**) right lateral view of specimen; (**B**) geometry of reconstructed endoskeletal elements; (**C**) complete reconstruction of right side of the specimen (slightly enlarged); paratype specimen (MCZ VPF-6504): (**D**) left dorsolateral view of the specimen; (**E**) geometry of reconstructed endoskeletal elements; (**F**) complete reconstruction in left laterodorsal view. Abbreviations: **asc**, ascending process of parasphenoid; **chy**, ceratohyal; **cl**, cleithrum; **clv**, clavicle; **dnt**, dentary; **exo**, exoccipitals; **hm**, hyomandibula; **mg**, median gular; **mx**, maxilla; **osph**, orbitosphenoid portion of braincase; **pal**, palatal complex; **pcl**, postcleithrum; **psp**, parasphenoid; **qd**, quadrate; **sc**, scales; **scl**, supracleithrum; **sy**, sympectic.

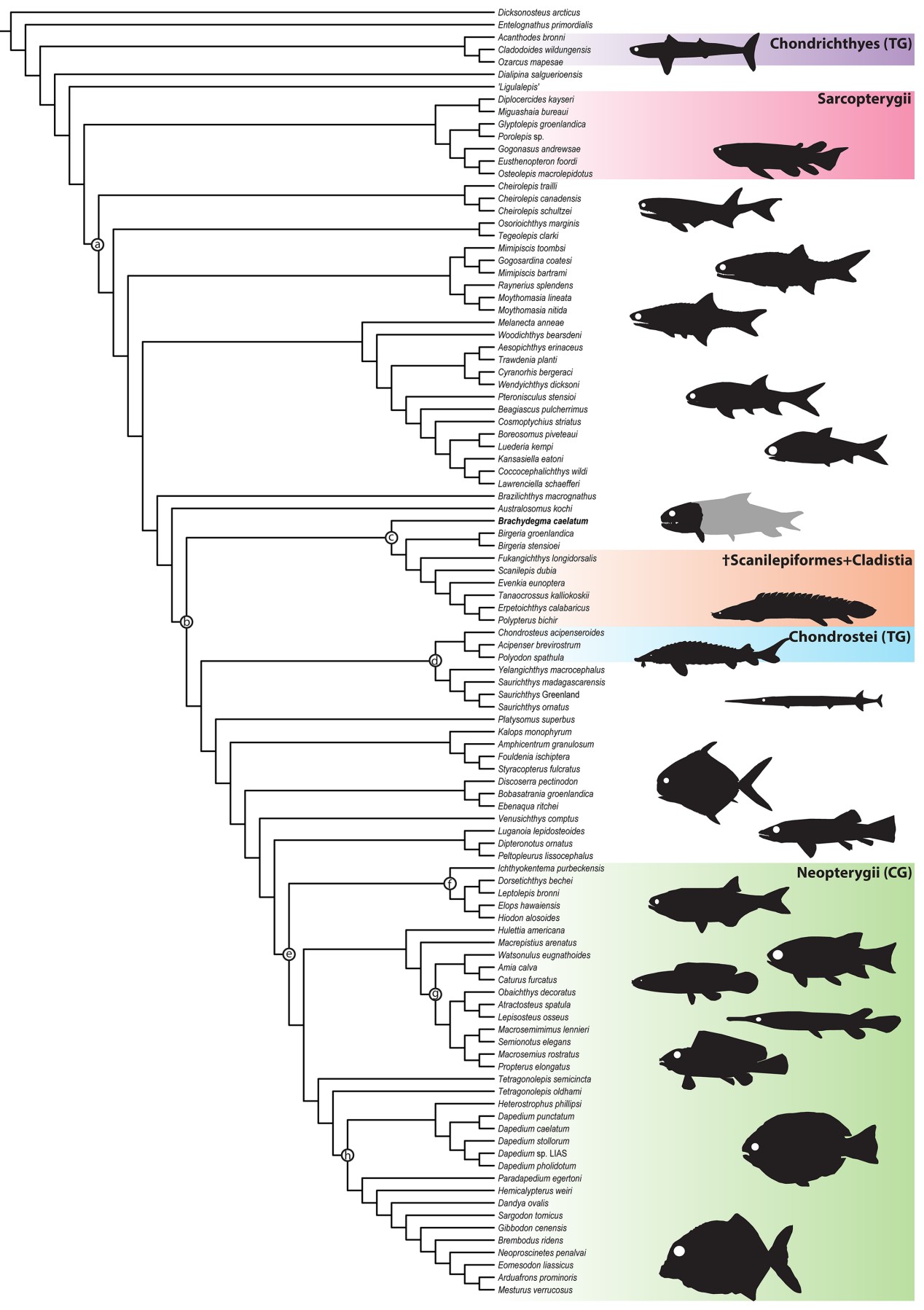

**Appendix 1—figure 3.** Agreement subtree (one of 23) resulting from parsimony analysis using equal weights and without constraining the topology of †*Brachydegma*. It contains 104 out of 117 taxa. Pruned taxa: †*Onychodus jandermarrai*; †*Styloichthys changei*; †*Guiyu oneiros*; †*Psarolepis romeri*; †*Meemania eos*; †*Donnrosenia schaefferi*; †*Howqualepis rostridens*; †*Moythomasia durgaringa*; †*Melanecta anneae*; †*Beishanichthys brevicaudalis*; †*Scopulipiscis saxciput*; †*Dapedium noricum;* †*Mesturus* sp. Lettered nodes as in text *Figure 11*.

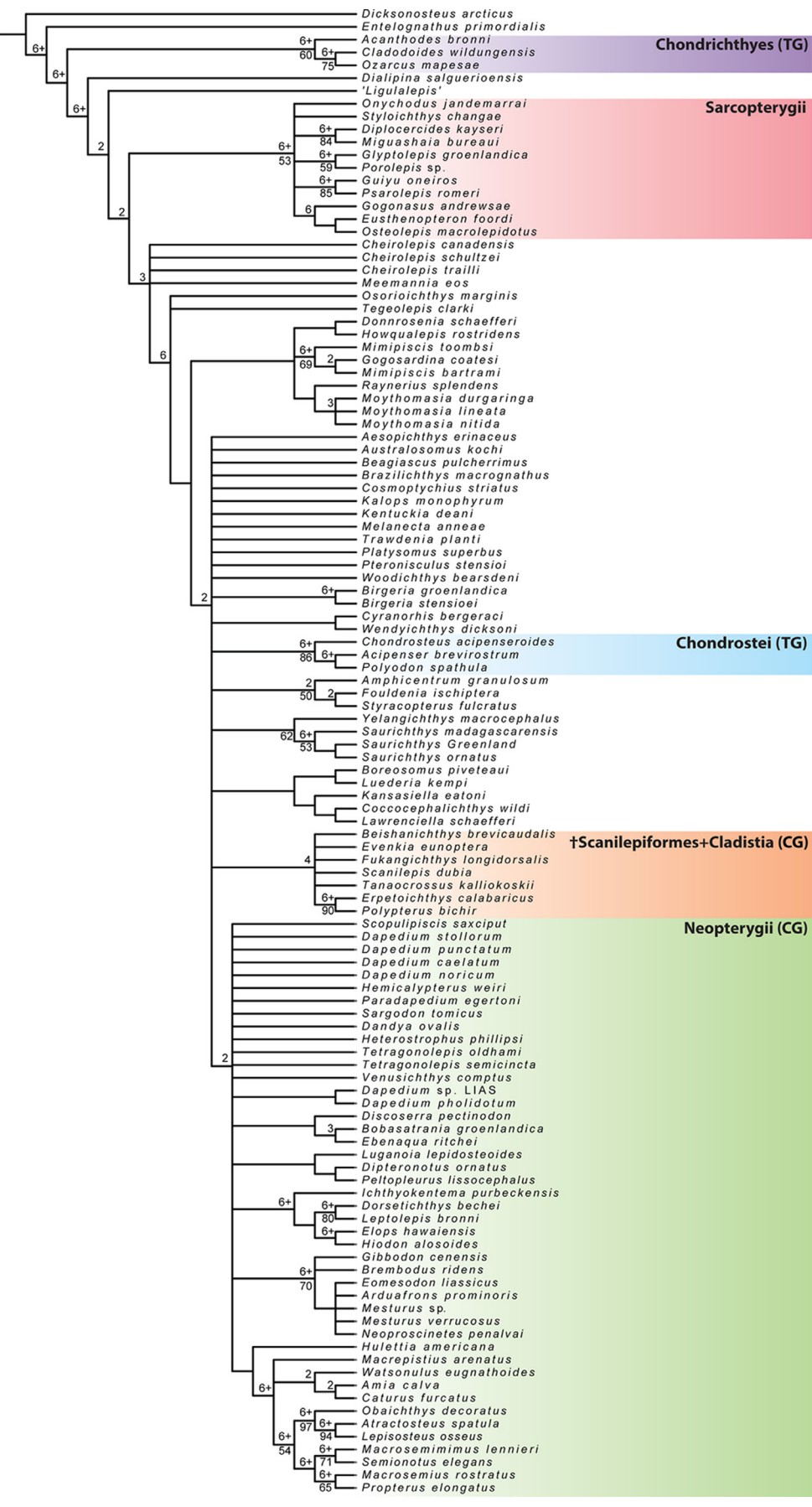

**Appendix 1—figure 4.** Strict consensus of the 1740 most parsimonious trees of 1646 steps for 116 taxa and 300 equally weighted characters. †*Brachydegma* was excluded. Consistency index =0.204, retention index =0.661. Numbers above nodes indicate Bremer values above 1. Numbers below nodes indicate bootstrap percentages above 50%.

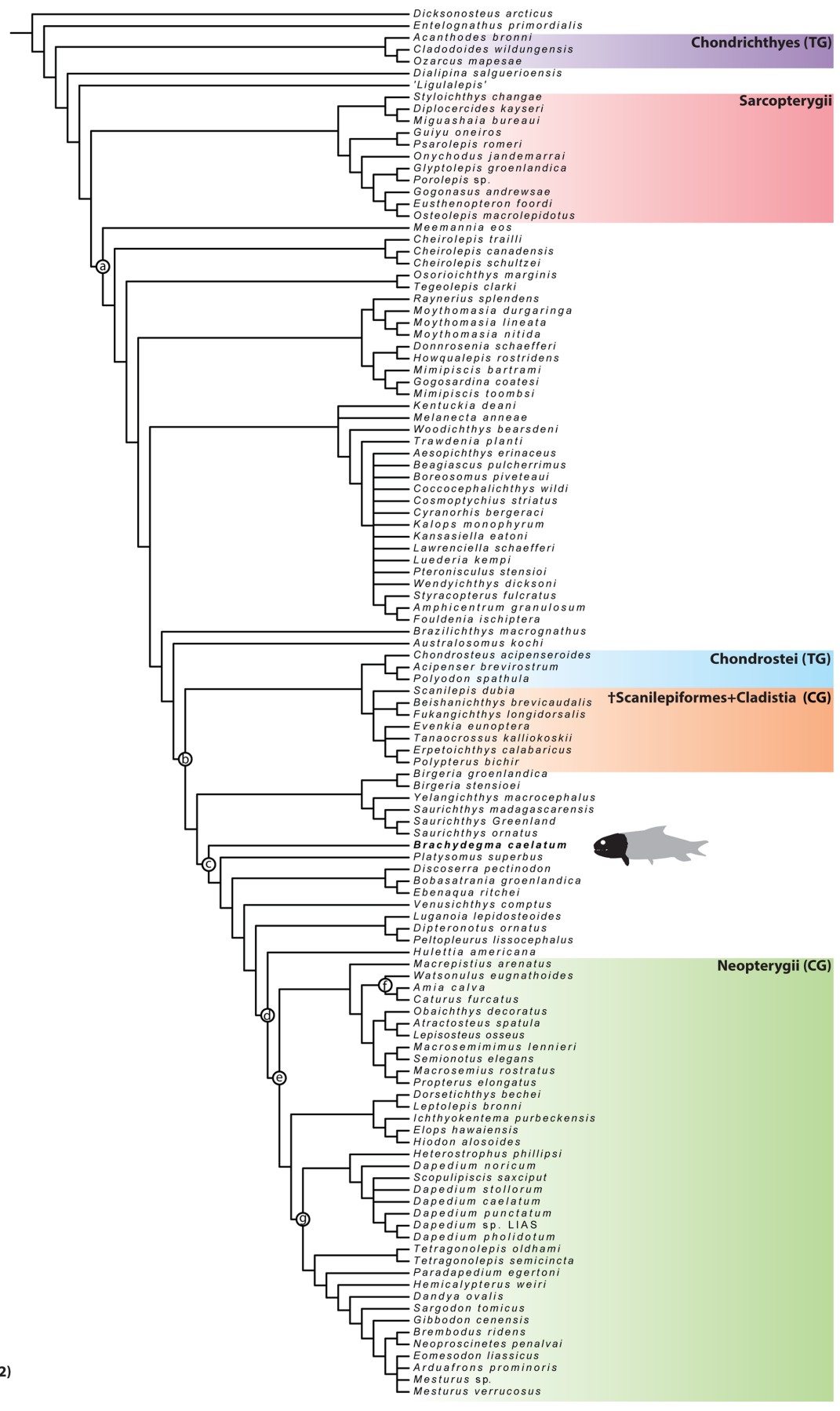

Implied weights (K=12)

**Appendix 1—figure 5.** Strict consensus of the 44 best fit trees (fit score =68.77) resulting from analyses using implied weights with a K=12. Selected node optimizations are as follows: (**a**) (Actinopterygii total group): C.43 (0→1); C.45 (1→0); C.139 (0→1); C.199 (0→1); (**b**) (Actinopterygii crown group): C.67 (1→0); C.101 (0→1); C.107 (0→1); C.157 (0→2); C.158 (0→1); C.174 (0→2); C.262 (0→1); (**c**) (†*Brachydegma* + Neopterygii total group): C.116 (0→1); C.124 (0→1); C.231 (0→1); (**d**) (†*Hulettia* + Neopterygii crown group): C56(0→1); C.73 (0→1); C.74 (0→1); C.90 (2→1); C.113 (0→1); C.119 (0→1); C.180 (0→1); C.219 (0→1); C.287 (0→1); (**e**) (Neopterygii crown group): C.18 (0→1); C.34 (0→1); C.149 (1→0); C.160 (0→1); C.171 (0→1); (**f**) (Halecomorphi + †*Watsonulus*): C.75 (0→1); C.76 (0→1); C.97 (0→1); C.135 (0→1); C.220 (1→0); C.280 (1→0); (**g**) (†Dapediidae + (†Pycnodontiformes + †*Tetragonolepis*)): C.28 (1→0); C.255 (0→2); C.257 (0→1); C.266 (0→1); C.278 (0→1); C.281 (0→1); C.284 (0→1); C.285 (0→1); C.293 (1→0).

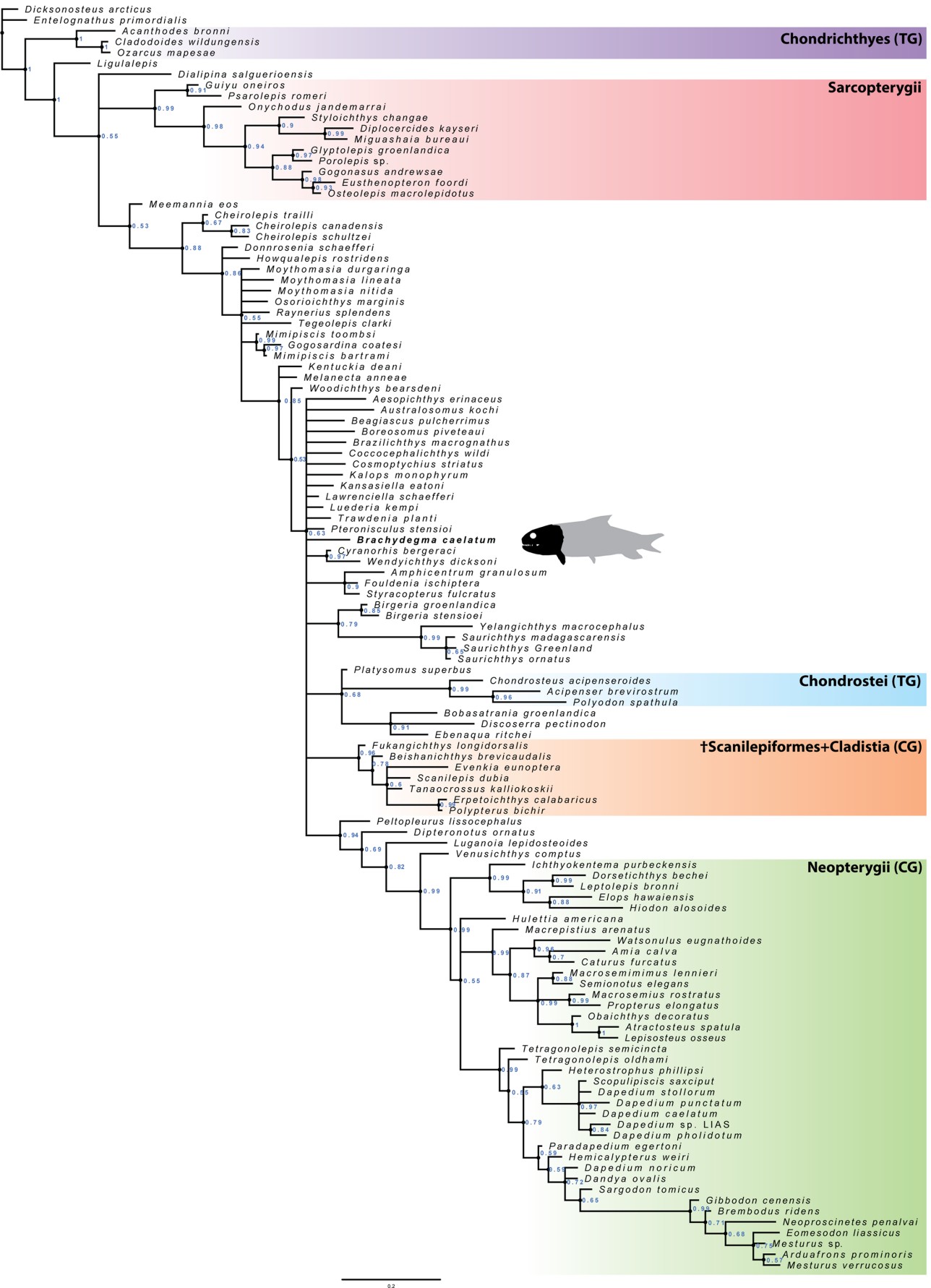

**Appendix 1—figure 6.** Phylogenetic tree from Bayesian analysis of morphological phylogenetic dataset. Values below nodes represent posterior probability support (BPP).

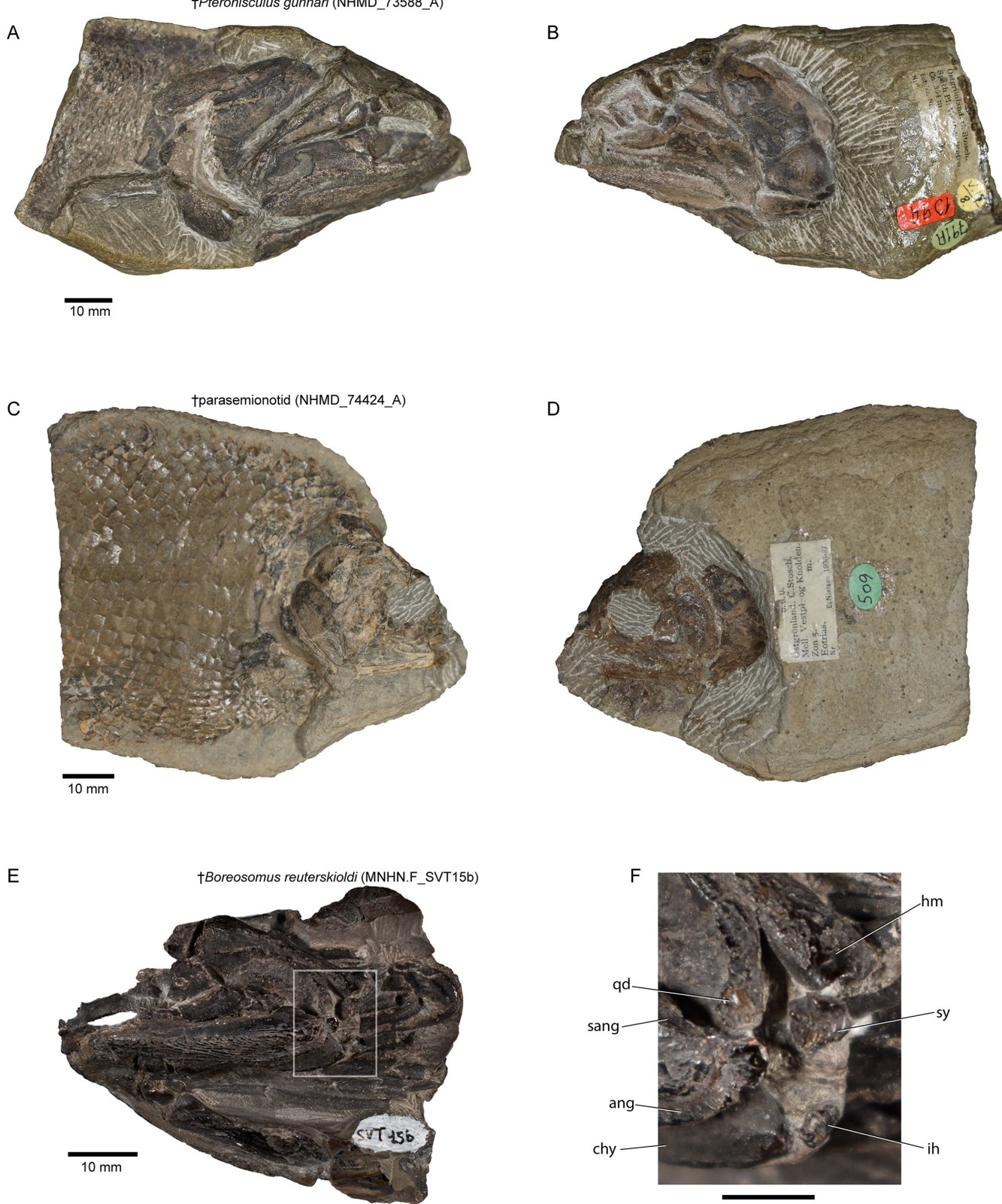

**Appendix 1—figure 7.** Triassic actinopterygians used for comparison with †*Brachydegma*. (**A**), (**B**) †*Pteronisculus gunnari* (NHMD_73588_A), Early Triassic, East Greenland; (**C**), (**D**) †Parasemionotidae indet. (NHMD_74424_A), Early Triassic, East Greenland; (**E**) †*Boreosomus reuterskioldi* (MNHN.F_ SVT15b), Early Triassic, Spitsbergen, Svalbard, prepared and figured by *Véran, 1988*; (**F**) magnification of jaw joint area, contained within a white box in (**E**). Abbreviations: **ang**, angular; **chy**, ceratohyal; **hm**, hyomandibula; **ih**, interhyal; **qd**, quadrate; **sang**, surangular; **sy**, symplectic. Scale bars equal 10 mm except for F where it equals 5 mm.

## Appendix 2

### Modifications and additions to morphological phylogenetic dataset

#### 1. General notes

We expanded the morphological character matrix of *Latimer and Giles, 2018*, by adding all taxa and most new characters, character states, and amendments presented in *Argyriou et al., 2018*, as well as adding and modifying existing characters (see detailed list of changes below regarding; *Argyriou et al., 2018*; *Latimer and Giles, 2018*). We removed a character regarding the presence of an ectopterygoid process on the palate (*Giles et al., 2017*), as the distribution of this feature needs to be reevaluated (see, for example, *Véran, 1996*). The external anatomy of †*Brachydegma*, which was included in a previous version of this matrix (*Giles et al., 2017*), was coded anew, while †*Saurichthys madagascariensis* was extensively recoded in *Argyriou et al., 2018*; these changes are not repeated here. We modified the scores for †*Trawdenia* (=†*Mesopoma*) *planti* sensu (*Coates and Tietjen, 2018*). We expanded our taxonomic coverage of stem ray-fins by adding †*Brazilichthys macrognathus* (*Figueroa et al., 2019*). Furthermore, to provide a better-informed picture of the distribution of accessory hyoidean elements in ray-fins, and also of neopterygian interrelationships, we scored the pycnodont †*Neoproscinetes penalvai*, for which both external (*Nursall and Maisey, 1987*) and braincase (*Machado, 2008*) anatomical information is available.

#### 2. List of added, removed, and modified characters

##### (A) Characters and character states added:

**C.21:** Both nostrils accommodated within single ossification: 0=absent, 1=present (from *Argyriou et al., 2018*).

**C.97:** Accessory hyoid element involvement in jaw joint: 0=absent, 1=present (modified from *Latimer and Giles, 2018*, and references therein). We modified the name of this character as well as the scores to reflect the uncertainty regarding the involvement of these rarely preserved elements in fossils. This character is coded only in taxa where accessory hyoid elements are present, or where there is adequate data regarding their presence, absence. For example, †*Pteronisculus stensioi* is coded as 1, following our investigations and interpretations (see also *Nielsen, 1942*).

**C.110:** Operculum: 0=absent, 1=present (from *Argyriou et al., 2018*).

**C.152:** Craniospinal process: 0=absent, 1=present (from *Gardiner et al., 2005*; *Argyriou et al., 2018*).

**C.159:** Birfurcation of dorsal aorta into lateral dorsal aortae: 0=open in endoskeletal groove, 1=enclosed in canal, 2=below parasphenoid (state 2 added, sensu *Argyriou et al., 2018*)

**C.167:** Occipital region ossification pattern: 0=basioccipital and exoccipitals as separate ossifications, 1=comineralized (from *Argyriou et al., 2018*).

**C.179:** Parasphenoid pierced by ascending common carotids: 0=absent, 1=present (from *Argyriou et al., 2018*).

**C.187:** Arrangement of olfactory nerve in orbital region: 0=completely enclosed in endoskeletal olfactory canal, 1=traversing the orbit lateral to the interorbital septum, at times leaving a groove on the latter (from *Argyriou et al., 2018*).

**C.201:** Lateral cranial canal connects to lateral wall of braincase: 0=absent, 1=present (from *Argyriou et al., 2018*).

**C.202:** Intramural diverticula opening in fossa bridgei: 0=absent, 1=present (from *Argyriou et al., 2018*).

**C.222:** Ossified accessory hyoid elements: 0=absent; 1=present (new character). In most primitive gnathostomes and chondrichthyans, elements situated between the epihyal or hyomandibula and the ceratohyal are completely absent.

**C.223:** If present, number of accessory hyoid elements: 0=one; 1=two (new character). The homology of some of the constituents of the hyoid arch – the so-called intermediate or accessory hyoid elements – in modern ray-fins remains controversial. The history attached to naming these elements is very complex (*Sewertzoff, 1928*; *Patterson, 1973*; *Patterson, 1982*; *Jollie, 1984*; *Véran, 1988*; *Gardiner and Schaeffer, 1989*; *Grande and Bemis, 1991*; *Gardiner et al., 1996*; *Grande and Bemis, 1998*; *Grande, 2010*; *Hilton et al., 2011*), and we have tried here to apply a simple, consistent approach and code for the number of ossified accessory hyoidean elements alone.

Polypterids exhibit a single ossified element connecting the hyomandibula with the ceratohyal, which has been identified as the primitively present interhyal (*Allis, 1922*; *Patterson, 1982*; *Jollie, 1984*). A second independently ossified element (a 'symplectic') does not develop (*Jollie, 1984*). Neopterygians (e.g., *Amia, Lepisosteus, Hiodon,* †*Dorsetichthys,* †*Macrosemionotus,* †*Watsonulus* etc.) exhibit two intermediate hyoidean elements, a symplectic and an interhyal. The interhyal connects the hyomandibula with the ceratohyal. This may be very reduced (e.g., †*Watsonulus, Elops*), or entirely cartilaginous (e.g., *Amia, Lepisosteus*) in many actinopts. The ossification that contacts the hyomandibula (and typically the quadrate and the articular), but does not articulate with the ceratohyal, is termed the symplectic. This element may brace the quadrate, and in †*Watsonulus,* †*Caturus,* Amia and possibly pycnodonts (*Patterson, 1973*; *Olsen, 1984*; *Véran, 1988*; *Gardiner et al., 1996*; *Grande and Bemis, 1998*), additionally forms an articulation with the lower jaw. Although the presence of a true symplectic in stem actinopterygians ('paleoniscoids') has been rejected by many authors (*Patterson, 1973*; *Patterson, 1982*; *Gardiner and Schaeffer, 1989*; *Gardiner et al., 1996*), Véran (*Véran, 1988*) identified a second, symplectic-like, intermediate ossification in the hyoid arch in a number of 'palaeoniscoids'. Our reexamination of part of her material (†*Boreosomus reuteskioldi*) confirmed her observations of two ossified accessory hyoidean elements. These include a symplectic-like element associated with the anteroventral portion of the hyomandibula and in close association with the palatoquadrate and lower jaw; and a small interhyal articulating with the posterodorsal tip of the ceratohyal. †*Brachydegma,* which is not a crown neopterygian, also exhibits both a symplectic and an interhyal. The condition in modern Chondrostei is extremely complicated. Two intermediate hyoidean elements are present, but unlike in other extant ray-fins these are serially arranged. Dorsally, a hypertrophied element connects the hyomandibula with the quadrate and lower jaw, and might partially ossify in very large sturgeons (*Hilton et al., 2011*). This element has been variably homologized with the symplectic or the interhyal. A second, smaller, element suspends the ventral portion of the hyoid arch from the former hypertrophied element, but does not ossify. This element has been identified as either an interhyal or a posterior ceratohyal. Given the fact that only one of the two elements ossify, we code *Acipenser* as 0.

**C.225:** Position of symplectic: 0=posterior to the posterior margin of quadrate, 1=medial to the posterior margin of quadrate (modified from *Arratia, 2013*).

**C.267:** Epineural processes: 0=absent, 1=present (from *Arratia, 2013*; *Argyriou et al., 2018*).

## (B) Character scores updated from *Latimer and Giles, 2018*, unless stated otherwise

†*Acantodes bronni*

 C.110:? → -
 C.111:? → -
 C.112:? → -
 C.113:? → -
 C.114:? → -

*Acipenser brevirostrum*

 C.111: 2 → -
 C.128: 1 → 0
 C.140: 0 → 1
 C.157: - → 1/2
 C.196:? → 1
 C.197:? → 0
 C.198:? → 1
 C.199:? → 0
 C.200: 0 → -
 C.207: - → 0
 C.208: - → 1
 C.209: - → 0
 C.211: - → 0
 C.212: - → 0
 C.215: 0 → 1
 C.219: 1 → 0

C.224: - →?
C.230: 0 → 0/1

*Amia calva*

C.157: 0 → 2

†*Amphicentrum granulosum*

C.72: 1 → 0

†*Arduafrons prominoris*

C.273: 0 → -

*Atractosteus spatula*

C.157: 0 → 2
C.186: 0 → 1

†*Australosomus kochi*

C.202: - → 0
C.224: - →?

†*Beishanichthys brevicaudalis*

C.224: - →?

†*Birgeria groenlandica*

C.3: 0 → 1
C.4: 0 → -
C.5: 1 → -
C.7: 0 → -
C.9: 0 → -
C.10: 0 → -
C.11: 1 → -
C.12: 0 → -
C.47: 0 → 1
C.64: 0 → 1
C.65: - →?
C.66: - → 1
C.70: 0 → 1
C.97: - →?
C.100: 2 → 0
C.111: 1 →?
C.157:? → 2
C.183: 1 →?
C.207: 1 → 0
C.209:? → 0
C.211:? → 0
C.219: 0 →?
C.224: - →?
C.244: - →?
C.245: - →?
C.249: 3 → 1
C.273: 0 → -

†*Birgeria stensioei* (changes regarding **Argyriou et al., 2018**)

C.270:? → 0
C.271:? → 0
C.272:? → 0
C.273:? → 0

†*Bobasatrania groenlandica*

C.224: - 1 →?

†*Boreosomus piveteaui*

C.188: 1 →?
C.224: - →?
C.273: 0 → -

†*Brembodus ridens*

C.273: 0 → -

†*Caturus furcatus*

C.273: 0 → -

†*Cheirolepis trailli*

C.224: - →?

†*Chondrosteus acipenseroides*

C.28: 1 → 0
C.52: 1 → 0
C.140: 0 →?
C.141: 0 →?

†*Dapedium* sp. (Lias)

C.201: - → 0
C.202: - →?

*Elops hawaiensis*

C.157: 0 → 2

†*Eomesodon liassicus*

C.273: 0 → -

†*Eusthenopteron foordi*

C.273: 0 → -

†*Fouldenia ischiptera*

C.43:? → 1

†*Fukangichthys longidorsalis*

C.221: 0 → 1
C.224: - →?

†*Gogonasus andrewsae*

C.152: 1 → 0

†*Hulettia americana*

C.157: 0 → 2
C.176: 1 → 2

†*Ichthyokentema purbeckensis*

C.157: 0 → 2

*Lepisosteus osseus*

C.157: 0 → 2

†*Leptolepis bronni*

C.154: 1 → 0
C.157: 0 → 2

C.273: 0 → -

†*Luederia kempi*

C.186:? → 0

†*Luganoia lepidosteoides*

C.52: 0 → 1
C.53: - → 0

†*Meemania eos*

C.224: - →?

†*Melanecta annae*

C.37: 1 →?

†*Mesturus verrucosus*

C.273: 0 → -

†*Mimipiscis bartrami*

C.273: 0 → -

†*Mimipiscis toombsi*

C.273: 0 → -

†*Moythomasia durgaringa*

C.273: 0 → -

†*Obaichthys decoratus*

C.157: 0 → 2

†*Ozarcus mapesae*

C.111:? → -
C.113:? → -
C.141: 0 → -
C.142: 0 → -

†*Peltopleurus lissocephalus*

C.69: 0 → 1
C.239: 1 →?
C.265: 0 → 1

*Polyodon spathula*

C.142: - → 0 (modified regarding *Argyriou et al., 2018*)
C.147:? → - (modified regarding *Argyriou et al., 2018*)
C.273: 0 → -

*Polypterus bichir*

C.153:? → 1
C.154:? → 0
C.155:? → 0
C.156:? → 1
C.157:? → 1

†*Pteronisculus stensioi*

C.224: - →?

†*Raynerius splendens*

C.201: 1 →?

C.224: - →?

†*Saurichthys madagascariensis*

C.7: maintained as 0 as in *Latimer and Giles, 2018*, and not *Argyriou et al., 2018*
C.14: 1 →?
C.20: 1 → -
C.22: 0 → -
C.62: maintained as 0 as in *Latimer and Giles, 2018*, and not *Argyriou et al., 2018*
C.64: maintained as 1 as in *Latimer and Giles, 2018*, and not *Argyriou et al., 2018*
C.65: maintained as 1 as in *Latimer and Giles, 2018*, and not *Argyriou et al., 2018*
C.66: 1 →?
C.101: 0 →?
C.111: - → 2
C.112: 1 → 0
C.113: - → 0
C.114: - → 0
C.131: 0 →?
C.136:? → 1
C.143: maintained as 0 as in *Latimer and Giles, 2018*, and not *Argyriou et al., 2018*
C.191: 1 →?
C.241: maintained as 0 as in *Latimer and Giles, 2018*, and not *Argyriou et al., 2018*
C.243: maintained as 1 as in *Latimer and Giles, 2018*, and not *Argyriou et al., 2018*
C.294: 0 → 1

†*Saurichthys* sp. Greenland (NHMD_157546_A)

C.119: 0 →? (modified regarding *Argyriou et al., 2018*)
C.142: - → 0 (modified regarding *Argyriou et al., 2018*)
C.198:? → 0 (modified regarding *Argyriou et al., 2018*)

†*Saurichthys ornatus*

C.142: - → 0
C.242: 0 →?
C.250:? → 1

†*Semionotus elegans*

C.174: 1 →?

†*Trawdenia* (=†*Mesopoma*) *planti*

C.33:? → 0
C.52:? → 1
C.54:? → 1
C.65:? → 0
C.71:? → 0
C.72:? → 0
C.76:? → 0
C.85:? → 2
C.90:? → 1
C.94:? → 1
C.95:? → 1
C.122:? → 1
C.186:? → 0
C.234:? → 1
C.235:? → 0
C.237:? → 0
C.240:? → 0
C.242:? → 1
C.243:? → 0
C.244:? → 1
C.245:? → 0
C.246:? → 0

C.247:? → 0
C.248:? → 1

†*Watsonulus eugnathoides*

C.112: 1 → 0
C.157: 0 → 2

