## [Editor Report]

This work is a valuable description and analysis of a fossil taxon with importance for understanding the complexities of actinopterygian evolution and is relevant to biologists and paleontologists interested in actinopterygians. Although in many ways *Brachydegma* raises more questions than it answers, the manuscript represents a substantive contribution to the subject – especially as *Brachydegma* has been thought by some researchers to occupy a quite different phylogenetic position and thus have a different significance to the overall story.

---

## [Decision Letter]

**Decision letter after peer review:**

Thank you for submitting your article "A Permian fish reveals widespread distribution of neopterygian feeding innovations" for consideration by *eLife*. Your article has been reviewed by 3 peer reviewers, including Min Zhu as Reviewing Editor and Reviewer #1, and the evaluation has been overseen by George Perry as the Senior Editor. The following individual involved in review of your submission has agreed to reveal their identity: Per E Ahlberg (Reviewer #2).

The reviewers have discussed the reviews with one another and the Reviewing Editor has drafted this decision to help you prepare a revised submission.

The homology and the evolution of the hyoid arch elements (accessory elements in particular) have impacted on the phylogeny of actinopterygians for decades. Whether the symplectic bones in halecomorphs, chondrosteids, coelacanths, and even in rhipidistians, are homologous or recurrently derived has been debated based on morphological, embryological and paleontological evidence. The lack of detailed morphological data on the hyoidean architecture from sufficient early actinopterygian taxa makes the whole story, which relates to the feeding mechanism of neopterygians, entangled. The major discovery reported here, from CT results, is the presence of what has been thought to be a neopterygian-specific hyoid arch arrangement, in which the intermediate bones (between the dorsal and ventral limbs) articulate in parallel rather than in series. As such, this work is a valuable description and analysis of a fossil taxon with above-average importance for understanding the complexities of actinopterygian evolution. Although in many ways Brachydegma raises more questions than it answers, the manuscript represents a substantive contribution to the subject – especially as Brachydegma has been thought by some researchers to occupy a quite different phylogenetic position and thus have a different significance to the overall story. However, the reviewers have some serious concerns on the present version for publication, even considering that the manuscript might, perhaps, have been submitted slightly too early.

Major comments from reviewers:

1) The quality of some of the segmentation (see detailed comments below). The reviewer would really like to see this rectified, something that will take a bit of time but can be done using the existing data sets and software.

2) The phylogenetic analysis places Brachydegma as a stem- polypterid, but this result is presented with caution. The authors state Brachydegma as a pivotal taxon of actinopterygians (l.79). Has this taxon any impact on the resolution of early actinopterygians. How is the phylogenetic result if we delete Brachydegma from the taxon list?

3) Abstract: Why do you use an imprecise and value-laden term such as "modern ray-finned fishes" in a context that is crying out for application of the precisely defined and unambiguous crown group / stem group / total group framework? What does "modern" mean here? How is the reader supposed to know whether you are using it phylogenetically as a synonym for "extant" or in a traditional progressivist evolutionary-grade sense of "advanced and implicitly superior"? If you are using it as a synonym of "extant", which seems most likely, then the two first sentences of the abstract can be re-expressed thus: "Molecular timescales indicate that the actinopterygian crown group (i.e. the clade containing all living ray-finned fishes) had originated by the end Carboniferous. However, the vast majority of late Paleozoic actinoptergyians appear to belong to the stem group, and conform to an anatomically conservative pattern little different from that of Devonian examples that are tens of millions of years older." This eliminates all ambiguity.

4) Lines 32-33: Again use of "modern". If you mean "crown neopterygian", please say so.

5) Line 50: Would it not have been appropriate to consider Rebecca Hitchin's PhD thesis on Acentrophorus in this paper? Admittedly there's little or no information about the endoskeleton, but the thing looks remarkably like a small semionotid and its Permian date makes it highly interesting in the present context.

6) Line 78 (and elsewhere): This seems like a good place to raise the problem of the figure quality – or rather, the quality of the segmentation (the actual rendering of the ply files from the segmentation process is fine). I have a good deal of personal hands-on experience of segmenting difficult data sets in Mimics, and I can see that you have been struggling with parts of the Brachydegma scans, but I would never consider a jagged outline such as that of the palatoquadrate complex in Figure 1E, the lower jaw in 1G, or the sphenoid region in 2E to be acceptable for publication. It looks like you have worked through each scan in only one view plane, without considering whether the result is plausible (or can be corrected, which it almost invariably can) in other planes. And what about the relationship between the articular (lavender) and prearticular (rusty brown) in 1G? We all know that the morphology you show here – isolated diagonal strips of prearticular on the articular – must be a segmentation artefact. Why have you not resolved this properly? Given that your entire paper rests on the accurate segmentation of your CT scans, I think it is important that you demonstrate maximum levels of care right across the morphology. For an example of what I mean, have a look at the segmentation of the new Devonian tetrapod jaw Brittagnathus in Ahlberg and Clack (2020, Royal Society Open Science). This was a horribly difficult scan with a killer combination of light ossification, low contrast, and 'blotchiness' caused by extensive mineral growth. It took me several months to segment and digitally dissect it, always switching back and forth between the three view planes to check everything. It should be possible to achieve results of at least equal quality with Brachydegma.

7) Line 138: A cross-section view of this phylogenetically important character would be valuable.

8) Line 179: Why do you refer to the teeth of the dentary as caniniform? The term doesn't mean 'big and pointy' in general, it means one or a few teeth that are substantially bigger than their neighbours in the tooth row, thus recalling the canines of predatory mammals. A whole tooth row by definition cannot be caniniform.

9) Line 191: Poor preservation I understand, but why should scan resolution be a factor here? The scapulocoracoid is a big structure and should easily be captured by a scan resolution good enough to reveal the symplectic and interhyal.

10) Line 228-229: I would have thought it a virtual certainty that a 'coronoid process' on the surangular is non-homologous with one on the prearticular. Apart from anything else, one is external to the jaw adductor musculature and the other internal to it. Why not code these states as non-homologous in the analysis?

11) Lines 443-459: This text refers several times to "resolution" when I think you actually mean "voxel size". The two terms are not synonymous: resolution is always lower than voxel size.

12) The organization is poor, with triplicates and duplicates of figures inflating the document and making the work difficult to navigate.

13) Don't give up! The prospects are good – comments here and below are intended as constructive.

14) The abstract is slightly off-target. What are the main, new, findings of this work and why are they significant?

15) The introduction needs a re-write. The primary discovery is the condition and significance of the hyoid arch. This topic needs a better, more fulsome, introduction – within the introduction! Use the intro. section to instruct readers in hyoid arch essentials: major conflicts in the literature (what literature?), and in hypotheses of homology: naming of the parts, consensus and conflict. The various tenuous hypotheses of Brachydegma's stem locations (from Gardiner and Schaeffer '89 onwards) could be summarized more effectively in a simple figure. The description is incomplete and needs more detail in places. Although many copies of figures are provided, important views of structures are missing. The analysis provides results but doesn't then examine or test these results, despite expressed concerns about the quality of these results. The discussion could be cut by half -noting that the biomechanical function(s) of neopterygian symplectic arrangements are not yet understood.

16) In short, the figures are almost done, the analyses are almost done, but the words are certainly not done. The figures aren't integrated properly with the text – especially in the far-flung but crucial appendix text on the dermal skeleton. The analysis relies heavily on characters of the dermal skeleton – why shove these details in the margins of the work as a whole?

---

## [Author Response]

Major comments from reviewers:1) The quality of some of the segmentation (see detailed comments below). The reviewer would really like to see this rectified, something that will take a bit of time but can be done using the existing data sets and software.

Both CT scans have been largely re-segmented and re-rendered and new images were produced.

2) The phylogenetic analysis places Brachydegma as a stem- polypterid, but this result is presented with caution. The authors state Brachydegma as a pivotal taxon of actinopterygians (l.79). Has this taxon any impact on the resolution of early actinopterygians. How is the phylogenetic result if we delete Brachydegma from the taxon list?

We have removed the word ‘pivotal’ from the text. Removing Brachydegma from the taxon list while keeping the newly added/modified characters and states results in a collapse of the actinopterygian crown-group, which is clearly explained in the text.

3) Abstract: Why do you use an imprecise and value-laden term such as "modern ray-finned fishes" in a context that is crying out for application of the precisely defined and unambiguous crown group / stem group / total group framework? What does "modern" mean here? How is the reader supposed to know whether you are using it phylogenetically as a synonym for "extant" or in a traditional progressivist evolutionary-grade sense of "advanced and implicitly superior"? If you are using it as a synonym of "extant", which seems most likely, then the two first sentences of the abstract can be re-expressed thus: "Molecular timescales indicate that the actinopterygian crown group (i.e. the clade containing all living ray-finned fishes) had originated by the end Carboniferous. However, the vast majority of late Paleozoic actinoptergyians appear to belong to the stem group, and conform to an anatomically conservative pattern little different from that of Devonian examples that are tens of millions of years older." This eliminates all ambiguity.

We agree with the reviewer and have edited the sentences as follows (additional edits made to keep the abstract within word limit): ‘The actinopterygian crown group (which comprises all living ray-finned fishes) originated by the end of the Carboniferous. However, most late Paleozoic taxa are stem actinopterygians, and broadly resemble stratigraphically older taxa.’

4) Lines 32-33: Again use of "modern". If you mean "crown neopterygian", please say so.

We have heavily revised this section in light of other comments and made sure to use ‘crown’ or ‘extant’ rather than ‘modern’ elsewhere in the text.

5) Line 50: Would it not have been appropriate to consider Rebecca Hitchin's PhD thesis on Acentrophorus in this paper? Admittedly there's little or no information about the endoskeleton, but the thing looks remarkably like a small semionotid and its Permian date makes it highly interesting in the present context.

We are reluctant to cite this thesis as it is part of the ‘grey’ literature and, furthermore, is not readily available onlline. However, we have added references to Patterson 1973, Gardiner 1960. We hope that planned work on *Acentrophorus* can clarify its anatomy and the interpretations of Hitchins.

6) Line 78 (and elsewhere): This seems like a good place to raise the problem of the figure quality – or rather, the quality of the segmentation (the actual rendering of the ply files from the segmentation process is fine). I have a good deal of personal hands-on experience of segmenting difficult data sets in Mimics, and I can see that you have been struggling with parts of the Brachydegma scans, but I would never consider a jagged outline such as that of the palatoquadrate complex in Figure 1E, the lower jaw in 1G, or the sphenoid region in 2E to be acceptable for publication. It looks like you have worked through each scan in only one view plane, without considering whether the result is plausible (or can be corrected, which it almost invariably can) in other planes. And what about the relationship between the articular (lavender) and prearticular (rusty brown) in 1G? We all know that the morphology you show here – isolated diagonal strips of prearticular on the articular – must be a segmentation artefact. Why have you not resolved this properly? Given that your entire paper rests on the accurate segmentation of your CT scans, I think it is important that you demonstrate maximum levels of care right across the morphology. For an example of what I mean, have a look at the segmentation of the new Devonian tetrapod jaw Brittagnathus in Ahlberg and Clack (2020, Royal Society Open Science). This was a horribly difficult scan with a killer combination of light ossification, low contrast, and 'blotchiness' caused by extensive mineral growth. It took me several months to segment and digitally dissect it, always switching back and forth between the three view planes to check everything. It should be possible to achieve results of at least equal quality with Brachydegma.

Segmentation of *Brachydegma* took many many months, and was halted when changes in circumstances meant that access to Mimics was made difficult. However, since first submitting the manuscript we have been able to carry out additional segmentation, and this is reflected in the re-rendered figures. We note that segmentation and interpretation of both tomographic datasets is particularly challenging, due to a series of factors, including low contrast and dense mineral growth in various portions of the scan.

7) Line 138: A cross-section view of this phylogenetically important character would be valuable.

We have added cross sections through the reconstructions of palate and maxilla to the figures to show the extensive connection.

8) Line 179: Why do you refer to the teeth of the dentary as caniniform? The term doesn't mean 'big and pointy' in general, it means one or a few teeth that are substantially bigger than their neighbours in the tooth row, thus recalling the canines of predatory mammals. A whole tooth row by definition cannot be caniniform.

We have rephrased this to ‘large, pointed teeth’.

9) Line 191: Poor preservation I understand, but why should scan resolution be a factor here? The scapulocoracoid is a big structure and should easily be captured by a scan resolution good enough to reveal the symplectic and interhyal.

Resolution was the wrong word to use. Preservation and scan penetration is highly variable across the specimen, and the scapulocoracoids were initially deemed too difficult and time-consuming to be segmented. However, renewed access to Mimics has meant that we were able to carry out this segmentation, and the figures and description have been updated accordingly.

10) Line 228-229: I would have thought it a virtual certainty that a 'coronoid process' on the surangular is non-homologous with one on the prearticular. Apart from anything else, one is external to the jaw adductor musculature and the other internal to it. Why not code these states as non-homologous in the analysis?

We agree with the reviewer that the coronoid process is a complicated structure that is likely to be non-homologous amongst many taxa. However, reflecting this in a character matrix is difficult. Our approach (and that of Giles et al., 2017 and subsequent analyses) is to include two characters concerning the coronoid process: one considering its absence/presence, and one contingent on presence with multiple states to allow for different anatomical conditions. At present, we cannot present an alternative way of coding this character, but we hope that more detailed work on the lower jaws of actinopterygians will provide a way to address this problem.

11) Lines 443-459: This text refers several times to "resolution" when I think you actually mean "voxel size". The two terms are not synonymous: resolution is always lower than voxel size.

We have replaced ‘resolution’ with ‘voxel size’.

12) The organization is poor, with triplicates and duplicates of figures inflating the document and making the work difficult to navigate.

We have increased the number of figures and reorganized them in order to increase the amount of anatomy figures and reduce duplicated panels

13) Don't give up! The prospects are good – comments here and below are intended as constructive.14) The abstract is slightly off-target. What are the main, new, findings of this work and why are they significant?

We have rewritten the abstract extensively.

15) The introduction needs a re-write. The primary discovery is the condition and significance of the hyoid arch. This topic needs a better, more fulsome, introduction – within the introduction! Use the intro. section to instruct readers in hyoid arch essentials: major conflicts in the literature (what literature?), and in hypotheses of homology: naming of the parts, consensus and conflict. The various tenuous hypotheses of Brachydegma's stem locations (from Gardiner and Schaeffer '89 onwards) could be summarized more effectively in a simple figure.

We have heavily revised the abstract in line with this and other comments. We now introduce the components of the hyoid arch in living and fossil actinopts, and explain the conflicting theories*.* We have also included a simplified figure showing the phylogenetic position of *Brachydegma* relative to a subset of taxa common to all previous phylogenetic analyses.

The description is incomplete and needs more detail in places. Although many copies of figures are provided, important views of structures are missing.

We have added more detail to the description and reorganized the figures in order to increase the amount of anatomy and views that are figured.

The analysis provides results but doesn't then examine or test these results, despite expressed concerns about the quality of these results.

We conducted a series of additional analyses to address the points raised by the reviewers: 1) analysis of the dataset after excluding *Brachydegma* from the taxon list; 2) analysis of the dataset after constraining *Brachydegma* with halecomorphs, as previously suggested (Hurley et al 2007); 3) analysis of the dataset after constraining *Brachydegma* with actinopterans (sensu Giles et al 2017); 4) parsimony analysis using implied weights, in order to evaluate an alternative placement, after allowing the algorithm to gently downweight homoplasic characters.

The discussion could be cut by half -noting that the biomechanical function(s) of neopterygian symplectic arrangements are not yet understood.

have shortened discussion and completely cut discussion of biomechanical functions

16) In short, the figures are almost done, the analyses are almost done, but the words are certainly not done. The figures aren't integrated properly with the text – especially in the far-flung but crucial appendix text on the dermal skeleton. The analysis relies heavily on characters of the dermal skeleton – why shove these details in the margins of the work as a whole?

Please see detailed comments elsewhere. We have revised the figures and analyses, and moved the dermal skeleton description to the main text. Moreover, we produced new detailed figures of the external anatomy of the type specimen (the exoskeleton of the paratype is not particularly informative, due to poor preservation), which are part of the main text.